# A tripartite synergistic optimization strategy for zinc-iodine batteries

Weibin Yan[1], Ying Liu ®[1,2] ✉, Jiazhen Qiu[1], Feipeng Tan[1], Jiahui Liang[1], Xinze Cai ®[1], Chunlong Dai[1,2], Jiangqi Zhao[1,2] ✉ & Zifeng Lin ®[1,2] ✉

The energy industry has taken notice of zinc-iodine (Zn-I$_2$) batteries for their high safety, low cost, and attractive energy density. However, the shuttling of I$_3^-$ by-products at cathode electrode and dendrite issues at Zn metal anode result in short cycle lifespan. Here, a tripartite synergistic optimization strategy is proposed, involving a MXene cathode host, a n-butanol electrolyte additive, and the in-situ solid electrolyte interface (SEI) protection. The MXene possesses catalytic ability to enhance the reaction kinetics and reduce I$_3^-$ by-products. Meanwhile, the partially dissolved n-butanol additive can work synergistically with MXene to inhibit the shuttling of I$_3^-$. Besides, the n-butanol and I$^-$ in the electrolyte can synergistically improve the solvation structure of Zn$^{2+}$. Moreover, an organic-inorganic hybrid SEI is in situ generated on the surface of the Zn anode, which induces stable non-dendritic zinc deposition. As a result, the fabricated batteries exhibit a high capacity of 0.30 mAh cm$^{-2}$ and a superior energy density of 0.34 mWh cm$^{-2}$ at a high specific current of 5 A g$^{-1}$ across 30,000 cycles, with a minimal capacity decay of 0.0004% per cycle. This work offers a promising strategy for the subsequent research to comprehensively improve battery performance.

To address the intermittent power generation of green energy sources such as solar and wind power, secondary battery systems stand out as mature energy storage devices for real-time dynamic storage[1]. Among the various secondary battery systems, aqueous zinc-ion batteries (AZIBs) have gained popularity due to low redox potential (−0.76 V vs. SHE), intrinsic safety, reasonable cost, natural abundance, and non-toxic nature[2,3]. However, the primary cathode materials, employing an intercalation/deintercalation-type storage mechanism such as vanadium oxide and manganese oxide, face rapid capacity decay due to structural collapse caused by the strong Coulombic interaction between embedded ions and surrounding lattices during the electrochemical process[4,5]. As a solution, iodine, the most abundant and least toxic halogen element, has been gradually introduced into AZIBs. The zinc-iodine (Zn-I$_2$) batteries operate through iodine/iodide ion conversion at a charge-recharge platform (1.38 V), exhibiting improved kinetics and smaller crystal structure dependence than the counterparts that operate the intercalation/deintercalation-type storage mechanism[6].

Nevertheless, iodine presents challenges such as poor conductivity, thermodynamic instability, and the generation of I$_3^-$ in the electrolyte due to the dissolution of cathode electrode materials. The shuttle effect of I$_3^-$ leads to rapid capacity decay and low Coulombic efficiency. To tackle these issues, the predominant strategy in designing iodine electrodes currently revolves around confining iodine to conductive substrate through physical or chemical confinement effects, such as Prussian blue analogs, various forms of carbon-based materials, MOFs and their derivatives, polymers. Among them, carbon-based carrier materials with high electrical conductivity and stability have become a good choice, given their porous structure that effectively restricts the movement of polyiodides. However, they rely solely on physical interactions constrained by the electrical neutrality and large pore size inherent in carbon-based carrier materials, leading to sluggish responses and low efficiency in constraining shuttle effects[7]. To overcome these limitations, MXenes, characterized by a general formula of M$_{n+1}$X$_n$T$_x$ (where M is a transition metal such as Ti,

[1]College of Materials Science and Engineering, Sichuan University, Chengdu, China. [2]Key Laboratory of Advanced Special Material & Technology, Ministry of Education, Chengdu, China. ✉e-mail: liuying5536@scu.edu.cn; jiangqizhao@scu.edu.cn; linzifeng@scu.edu.cn

Nb, Mo, etc., X represents C or N, and $T_x$ signifies surface terminations like -OH, =O, -F, etc.), emerge as a promising alternative to carbon materials for loading iodine. MXenes boast high conductivity, dense surface functional terminations, a distinctive stratified structure, and a tunable bandgap[8]. These unique properties enable MXenes to establish a robust connection with iodine and its reaction products, effectively anchoring iodine and inhibiting dissolution to suppress shuttle effects[9].

On the other hand, high-performance zinc anode electrodes are essential for the construction of advanced Zn-I₂ batteries. Currently, intensive efforts are devoted to protecting the zinc anode electrode from the growth of zinc dendrites, hydrogen evolution reaction (HER), and corrosion[10,11]. And these protection strategies can be broadly categorized into four directions, including anode electrode material construction, surface passivation through coating, membrane modification, and electrolyte optimization. Among them, electrolyte optimization is the most convenient and effective solution to protect the zinc anode. Especially organic additives such as N-methylpyrrolidone (NMP)[12], sulfolane (SL)[13], ethylene glycol (EG)[14], and ethanesulfonamide (ESA)[15] have been identified for their ability to replace water molecules in the solvation sheath of $Zn^{2+}$. N-butanol, as a short-chain organic molecule, is more easily involved in solvation structures compared to other organic molecules. At the same time, the alkyl chains and hydroxyl groups of n-butanol are electron-donating groups, making it easier to disrupt the hydrogen bond network in the solution and undergo the desolvation processes. The superior zinc affinity and hydrophobicity of n-butanol make it easier to adsorb on the surface of zinc and mitigate side reactions[15,16].

While the current reports have made some progress in improving performance of Zn-I₂ batteries, the majority of current research primarily concentrates on enhancing either the cathode electrode, anode electrode, or electrolyte, often overlooking the importance of addressing the other part. As a result, this oversight can lead to an imbalance between capacity and cycle life, resulting in unsatisfactory overall performance of the Zn-I₂ batteries and significantly impeding their practical application[2,3,6]. Therefore, there is an urgent need for the comprehensive improvement of Zn-I₂ batteries from multiple perspectives.

In this study, we successfully employed a tripartite synergistic optimization strategy to achieve a practical Zn-I₂ batteries that own high capacity, high energy density, and prolong cycle life. We discover that the two-dimensional MXene cathode host with rich surface terminations and the suspension electrolyte formed by introducing n-butanol can synergistically limit the movement of $I_3^-$ towards the anode electrode. Simultaneously, the terminal metal sites exposed at the high conductivity MXene play a catalytic role in the conversion reaction of iodine. Based on molecular dynamic (MD) simulations, $I^-$ and n-butanol in the electrolyte participates in the solvation structure, reducing the free water content and aiding in the desolvation process. More importantly, n-butanol preferentially adsorbs on the surface of zinc anode electrode to form an in-situ film to prevent dendritic growth and induce uniform deposition of zinc along the (002) plane. Consequently, the fabricated full batteries exhibit a long cycle life of 30,000 cycles with a high capacity of 0.30 mAh cm⁻² and a superior energy density of 0.34 mWh cm⁻² at 5 A g⁻¹ with a high capacity retention of 88.23%, surpassing most other reported Zn-I₂ batteries. Our work provides another approach for comprehensively improving the capacity performance and lifespan of aqueous Zn-I₂ batteries and may become potential candidates for future large-scale energy storage devices.

## Results

The design of the tripartite synergistic optimization strategy is illustrated in Fig. 1, which includes three parts: 1) MXene (TMX) and suspended n-butanol work synergistically in the cathode region. The 2D transition metal carbides TMX cathode host with layered structure and rich surface groups provide abundant sites for $I^-/I_2$ conversion reaction and accelerate the conversion reaction that leads to the reducing of the formation of $I_3^-$, whose movement can be further effectively hindered by chemical restriction of TMX. Meanwhile, the suspended n-butanol formed on the cathode side due to its low solubility in aqueous solution can further limit the shuttle of free $I_3^-$ that has not been effectively adsorbed by TMX owing to its strong adsorption

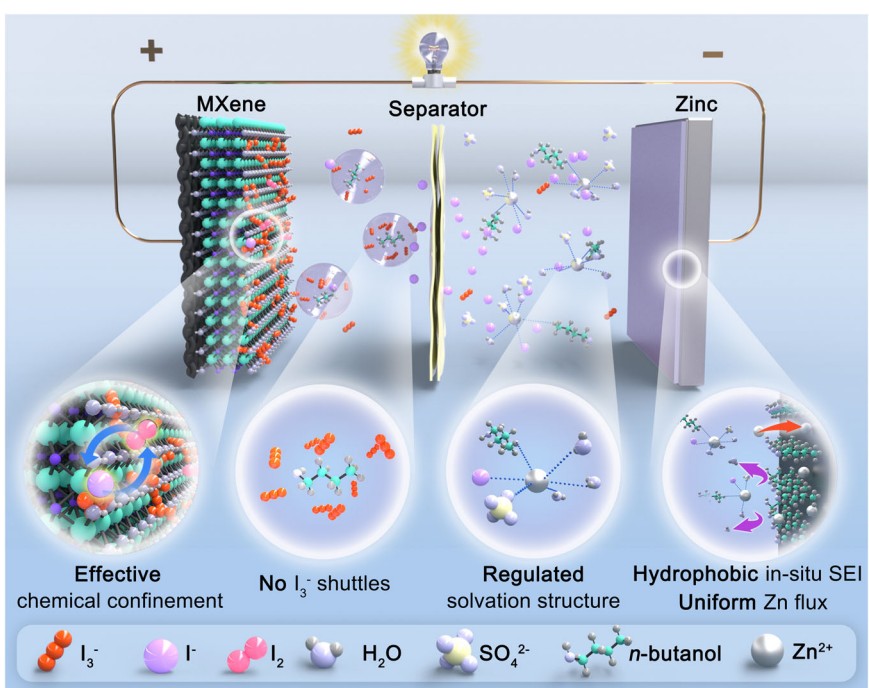

**Fig. 1 | Design principle of the Zn-I₂ battery.** Schematic illustration of the tripartite synergistic optimization strategy with cathode host, electrolyte additive, and in-situ anode protection layer.

capacity for $I_3^-$. 2) $I^-$ and n-butanol in the electrolyte synergistically regulate the solvation structure. $ZnI_2$, as a part of the original electrolyte, becomes the source of iodine reaction. Simultaneously, it actively participates in the solvation structure and displaces a portion of active water, which can effectively suppress the formation of dendrites. Furthermore, the electrolyte additive, n-butanol, also enters the solvated shell layer to reduce the content of active water. This optimized $Zn^{2+}$ coordination environment creates a highly stable interfacial condition for zinc plating and stripping. 3) adsorbed n-butanol on zinc surface and in-situ solid electrolyte interface (SEI) synergistically protect zinc anode. n-butanol will preferentially adsorb on the surface of zinc compared to other ions or molecules in the electrolyte, which is evidenced by the calculation of adsorption energy through DFT. The surface-bound n-butanol induces zinc growth along the (002) crystal plane and promotes a flat surface. Moreover, n-butanol can form hydrophobic SEI with certain inorganic substances due to the hydrophobicity of n-butanol, which prevents zinc dendrite formation and improves the cycle life.

### The function of Ti$_3$C$_2$T$_x$ MXene cathode host

The TMX was synthesized using the common etching method as described in experimental section[17]. The successful removal of the aluminum layer is evidenced by the disappearance of the diffraction peaks of $Ti_3AlC_2$ and appearance of the (002) peak of TMX at low angle in X-ray diffraction (XRD) pattern (Supplementary Fig. 1)[8]. Additionally, the TMX exhibits fewer and less dense peaks compared to $Ti_3AlC_2$, indicating that TMX owns a thinner layered structure[18]. Its thin-layer structure (thickness of 1.9 nm) is confirmed by scanning electron microscopy (SEM) and atomic force microscopy (AFM) (Fig. 2a and b), consistent with previous reports[19]. Besides, elemental mapping in Supplementary Fig. 2 indicates the uniform distribution of Ti, O, F, and C on TMX sheets.

To investigate impact of TMX on $I_3^-$ adsorption, fresh carbon cloth (CC) and CC coated with TMX were immersed in $I_3^-/H_2O$ solution. After a week of soaking, UV spectrum results of these two solutions (Supplementary Fig. 3) indicate significantly weaker $I_3^-$ bands for the solution treated with TMX compared to that treated with CC, showcasing the better adsorption effect of TMX on $I_3^-$[20]. Further UV testing on fully charged solutions at 1.8 V reveals a more pronounced concentration difference between the electrolytes with TMX and CC as cathode electrodes (Fig. 2c), which confirms the adsorption ability of TMX for $I_3^-$[21]. Moreover, we conducted theoretical calculations to assess adsorption ability, the variance in charge density, and the corresponding changes in intermolecular length. The simulated binding energy identifies a significantly lower value for TMX (−2.90 eV) compared to CC (−0.25 eV), indicating the thermodynamic advantage of TMX for $I_3^-$ adsorption (Fig. 2d)[22]. Besides, analyzing the charge density difference and corresponding intermolecular length provides a more profound insight into the reasons for the discrepancy in $I_3^-$ adsorption on different substrates. As depicted in Fig. 2e and f (the yellow areas indicate electron aggregation and the blue areas signify dissipation), the strong interaction between TMX and $I_3^-$ proves beneficial for restricting $I_3^-$ escape and reducing side reactions resulting from $I_3^-$ migration[23,24]. In addition, the mechanism of iodine conversion reaction on TMX is described in Note 1 and Supplementary Figs. 4–7.

Then Cyclic voltammetry (CV) results were initially employed to characterize electrochemical performance in a two-electrode system (TMX as cathode host and zinc as anode) with a mixture electrolyte of 2 M $ZnSO_4$ and 0.2 M $ZnI_2$ (named ZSI), ranging from 0.2 to 1.8 V at a scan rate of 0.3 mV s$^{-1}$ (Fig. 2g). It should be noted that the concentration of $ZnI_2$ (0.2 M) was selected through experimental exploration, and the corresponding discussion and data are presented in Note 2 and Supplementary Figs. 8–9. Both TMX and CC exhibit redox peaks at 1.41/1.28 V and 1.42/1.26 V, respectively, suggesting the conversion of $I^-/I_2$ during charge and discharge process.

It is worth noting that TMX demonstrates higher peak-specific current and a larger integral area, which indicates a higher capacity of TMX. This superiority is attributed to the large interlayer structure for fast ion transport and abundant surface functional groups of TMX for inhibiting the movement of $I_3^-$. Notably, TMX exhibits a distinct low polarization voltage difference around 20.0 mV, enhancing redox kinetics and reversibility stemming from the unique electronic structure and interaction between host materials and iodine species compared to CC. The Tafel slope (η) of TMX cathode host, derived from CV curves, is also smaller in both reduction and oxidation processes (179.2 and 58.9 mV dec$^{-1}$) than that of CC (197.7 and 70.8 mV dec$^{-1}$). A smaller η signifies higher electrocatalytic activity and faster thermodynamic conversion kinetics[25,26], which is also confirmed by CV measurements at various scan rates (Supplementary Fig. 10) and the electrochemical impedance spectroscopy (EIS) profiles (Supplementary Fig. 11)[27]. Figure 2i and Supplementary Fig. 12 present the galvanostatic charge-discharge (GCD) curves at various large current densities of 5, 10, 15, and 20 A g$^{-1}$, respectively. TMX appears a discharge capacity of 0.29 mAh cm$^{-2}$ and a charge capacity of 0.32 mAh cm$^{-2}$ at a specific current of 5 A g$^{-1}$ with a high Coulombic efficiency of 91%. In contrast, CC exhibits a lower charge capacity of 0.18 mAh cm$^{-2}$ and its unsatisfactory discharge capacity (0.16 mAh cm$^{-2}$) with an inferior Coulombic efficiency of 88%. The high capacity and Coulombic efficiency indicate that the large interlayer structure and unique electronic structure of TMX cathode host provide more reaction sites and faster reaction kinetics for $I_2/I^-$. The corresponding rate capacities follow a similar trend (Fig. 2j). At the same specific current, the capacity of TMX consistently surpasses that of CC. Even under an ultrahigh specific current of 20 A g$^{-1}$, TMX still achieves a substantial capacity of 0.19 mAh cm$^{-2}$, in stark contrast to the meager capacity of CC (0.08 mAh cm$^{-2}$ only). In addition, for every 1 A g$^{-1}$ increase in specific current, the capacity retention rate of TMX only decreases by 2%.

The long-term stability under constant specific currents has become a crucial indicator for assessing batteries. In the case of Zn-$I_2$ batteries, stable long cycles signify the capability of electrode material to immobilize iodine and inhibit the formation or shuttle of $I_3^-$. In this context, the Zn-$I_2$ battery, utilizing TMX as the cathode host, demonstrates a high reversible capacity of 0.22 mAh cm$^{-2}$ after 30,000 cycles at a high specific current of 5 A g$^{-1}$ with low capacity decay around 0.0012% per cycle (Fig. 2k). In contrast, the CC exhibits only 0.08 mAh cm$^{-2}$ after 6000 cycles at the same specific current and the capacity decay is up to 0.005% per cycle. It greatly confirms the feasibility of TMX as a cathode host.

### Characterization and simulation of electrolyte structure

The performance in the previous section is not mediocre, but it remains relatively unsatisfactory, particularly in terms of long-cycle performance[28]. Therefore, we further analyze and improve the structure of electrolytes. According to previous reports on electrolytes, the introduction of $I^-$ in ZSI aids in zinc deposition and accelerates reaction kinetics by involving $I^-$ in the solvation structure, as compared to the $ZnSO_4$ solution (ZS)[29]. Furthermore, numerous alcohols serve as electrolyte additives to alter the solvation structure and enhance the lifespan of zinc anode electrodes (as shown in Table S1)[14,30], but they are powerless to solve the problem of $I_3^-$ shuttle. Therefore, we are committed to exploring electrolyte additives that can simultaneously solve these problems. Ultimately, n-butanol is incorporated into the ZSI electrolyte (named ZSI-n) in this study, and the optimal amount (3% v/v) of n-butanol is determined through experiments (the corresponding discussion and data are included in the Note 3 and Supplementary Figs. 13–15). As shown in Supplementary Fig. 16, the weak $I_3^-$ bands of ZSI-n electrolyte are observed in UV-vis spectra. In contrast, the extra $I_3^-$ bands, and $I_5^-$ bands are shown in Raman spectra for ZSI electrolytes (Supplementary Fig. 17). These results reveal that the introduction of

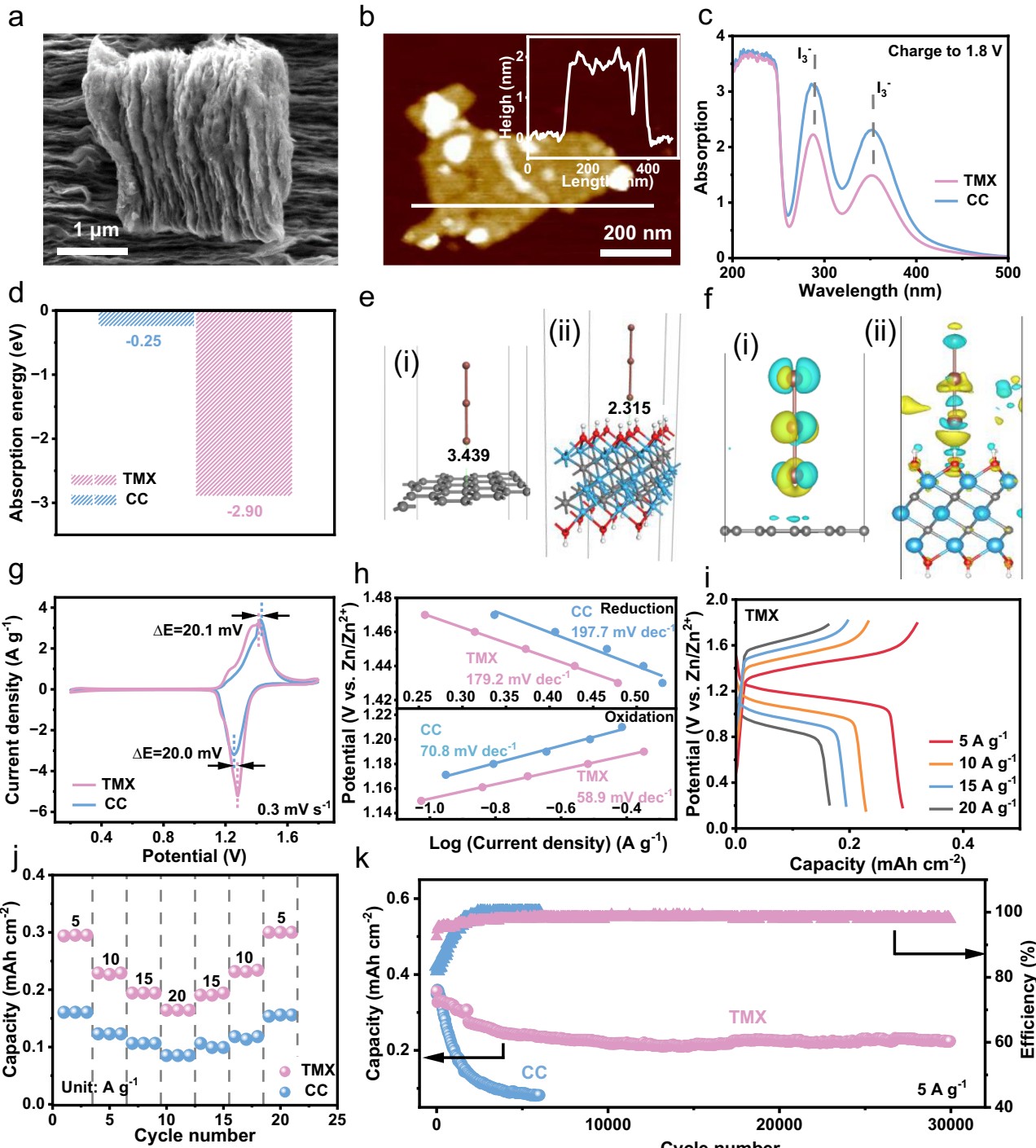

**Fig. 2 | Morphological and electrochemical performance of Ti₃C₂Tₓ MXene cathode host. a** SEM, and (**b**) AFM images of TMX. **c** UV-vis spectra of ZSI electrolyte with TMX and CC cathode host after charging to 1.8 V. **d** Values of surface adsorption energy of TMX and CC toward $I_3^-$. **e** The intermolecular length changes of (i) CC and (ii) TMX to absorb $I_3^-$. **f** The corresponding contour plots of the difference in charge density for (i) CC and (ii) TMX to absorb $I_3^-$. Light blue and yellow color indicate charge depletion and accumulation, respectively. **g** CV curves of TMX and CC samples at 0.2 mV s⁻¹. **h** Tafel slope of the initial discharge stage and charge stage. **i** Galvanostatic charge-discharge (GCD) curves of TMX. **j** Rate performance comparison of TMX and CC. **k** Capacity and Coulombic efficiency of TMX and CC at 5 A g⁻¹.

n-butanol limits the movement of $I_3^-$ [2,21,31]. In order to more intuitively reflect this role of n-butanol, the mechanism of $I_3^-$ shuttle is studied and discussed in Note 4 and Supplementary Figs. 4–5. Additionally, we added n-butanol to a solution filled with $I_3^-$ and measured the UV-vis spectrum after standing for 72 hours. It can be observed that the peak intensity of $I_3^-$ in the solution without adding n-butanol is only slightly lower, while in the solution with n-butanol, only a weak peak

of $I_3^-$ exists (Supplementary Fig. 18), which demonstrates the better adsorption capacity of n-butanol for $I_3^-$, further substantiating the above conclusion. The excellent ability of n-butanol to inhibit $I_3^-$ shuttle is not only due to its strong adsorption affinity with $I_3^-$, but also because it inhibits iodine dissolution, thereby further reducing the formation of $I_3^-$. The appeal statement is further confirmed by ex-situ UV-vis and Raman spectroscopy, and the corresponding

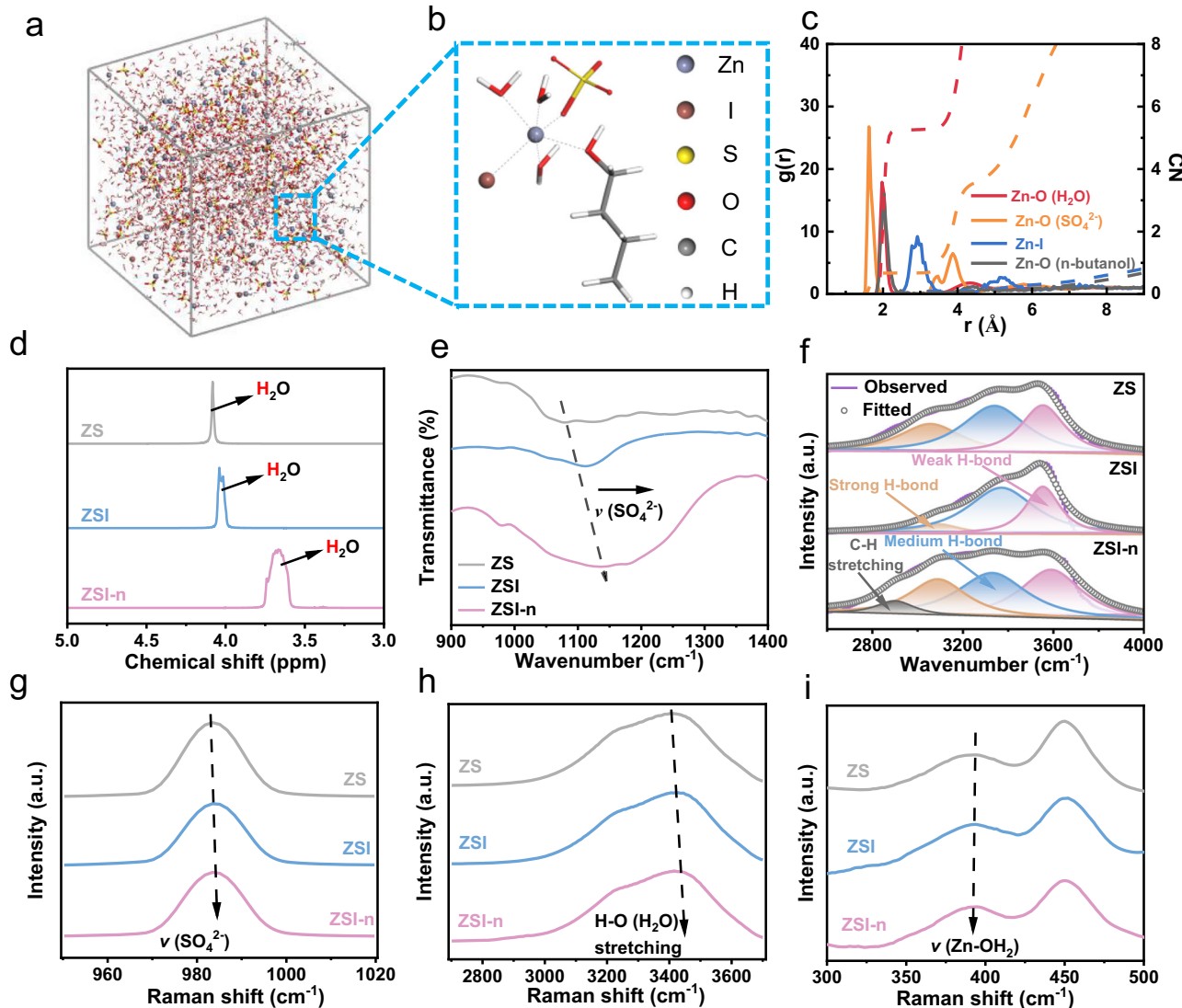

**Fig. 3 | Characterization and simulation of electrolyte. a** Snapshot of MD simulation box for ZSI-n. **b** enlarger view of snapshot for ZSI-n. **c** RDFs for coordinated bond with $Zn^{2+}$ in ZSI-n. **d** $^1H$ NMR spectra for ZS, ZSI and ZSI-n. **e** FTIR of three electrolytes between 900 and 1400 $cm^{-1}$. **f** Fitted FTIR of three electrolytes. Raman spectra between (**g**) 2600-3800 $cm^{-1}$, (**h**) 950–1020 $cm^{-1}$, and (**i**) 300–500 $cm^{-1}$.

discussions and data are shown in Note 5, Supplementary Fig. 5 and 19–21.

To investigate alterations in the solvation structure, the outcomes of molecular dynamics (MD) simulations were presented (Fig. 3a, Fig. 3b and Supplementary Fig. 22) to illustrate the clear involvement of $I^-$ and n-butanol in the composition of the solvation structure. Moreover, radial distribution functions (RDFs) simulated the distribution of molecules around the reference $Zn^{2+}$. The two sharp peaks around 0.2 nm in Supplementary Fig. 23a represent the $Zn^{2+}$-O ($H_2O$) and $Zn^{2+}$-O ($SO_4^{2-}$) interactions in pristine 2 M $ZnSO_4$ electrolyte, respectively. With the addition of $I^-$, a new Zn-I coordination structure appeared at the peak of 0.30 nm (Supplementary Fig. 23b), with a coordination number N(r) of 0.14. As a result, the N(r) of $Zn^{2+}$-O ($SO_4^{2-}$) changes significantly from 0.63 to 0.53, while the coordination number of water decreases by 0.1. The RDF of ZSI-n electrolyte in Fig. 3c indicates a further lower N(r) of $Zn^{2+}$-O ($H_2O$) from 5.35 to 5.26 while a clear peak of $Zn^{2+}$-O (n-butanol) is observed at about 0.20 nm and the N(r) of 0.11. Such variation trend indicates the decreasing of solvent water with the participation of $I^-$ and n-butanol.

$^1H$ nuclear magnetic resonance (NMR) spectroscopy (Fig. 3d) was further employed to examine the impact of electrolyte additives on

$H_2O$ molecules. Clearly, the addition of $I^-$ induces a shift in the $^1H$ peak towards a lower chemical shift, signifying an increase in the electron density of water protons and a weakening of H-bonds. Moreover, the introduction of n-butanol causes the peak to shift with a more pronounced span. This shift is attributed to strong dipole-dipole interactions between n-butanol and water, occurring not only in the solvation sheath of $Zn^{2+}$ but also between other particles outside the sheath. This indicates that n-butanol is more likely to donate electrons to water protons, resulting in the breakage of H-bonds and stronger O-H bonds in ZSI-n compared to that in ZSI. This phenomenon is supported by Fourier transform infrared (FTIR) spectra, which sensitively capture changes in molecular dipole moments. The $v(SO_4^{2-})$ stretching in Fig. 3e, located at -1079.2 $cm^{-1}$ in 2 M $ZnSO_4$ solution, exhibits the same offset direction in other solutions, confirming changes in solvation structure. As shown in Fig. 3f, the O-H stretching vibrations reflecting H-bond networks are categorized into weak H-bonds (3085 $cm^{-1}$), medium H-bonds (3338 $cm^{-1}$), and strong H-bonds (3554 $cm^{-1}$). $I^-$ cause the rupture of many strong H-bonds, leading to the formation of medium H-bonds. Further addition of n-butanol causes the transformation of some medium H-bonds into weak H-bonds. The heightened peak of strong H-bonds in the ZSI-n solution is attributed to the

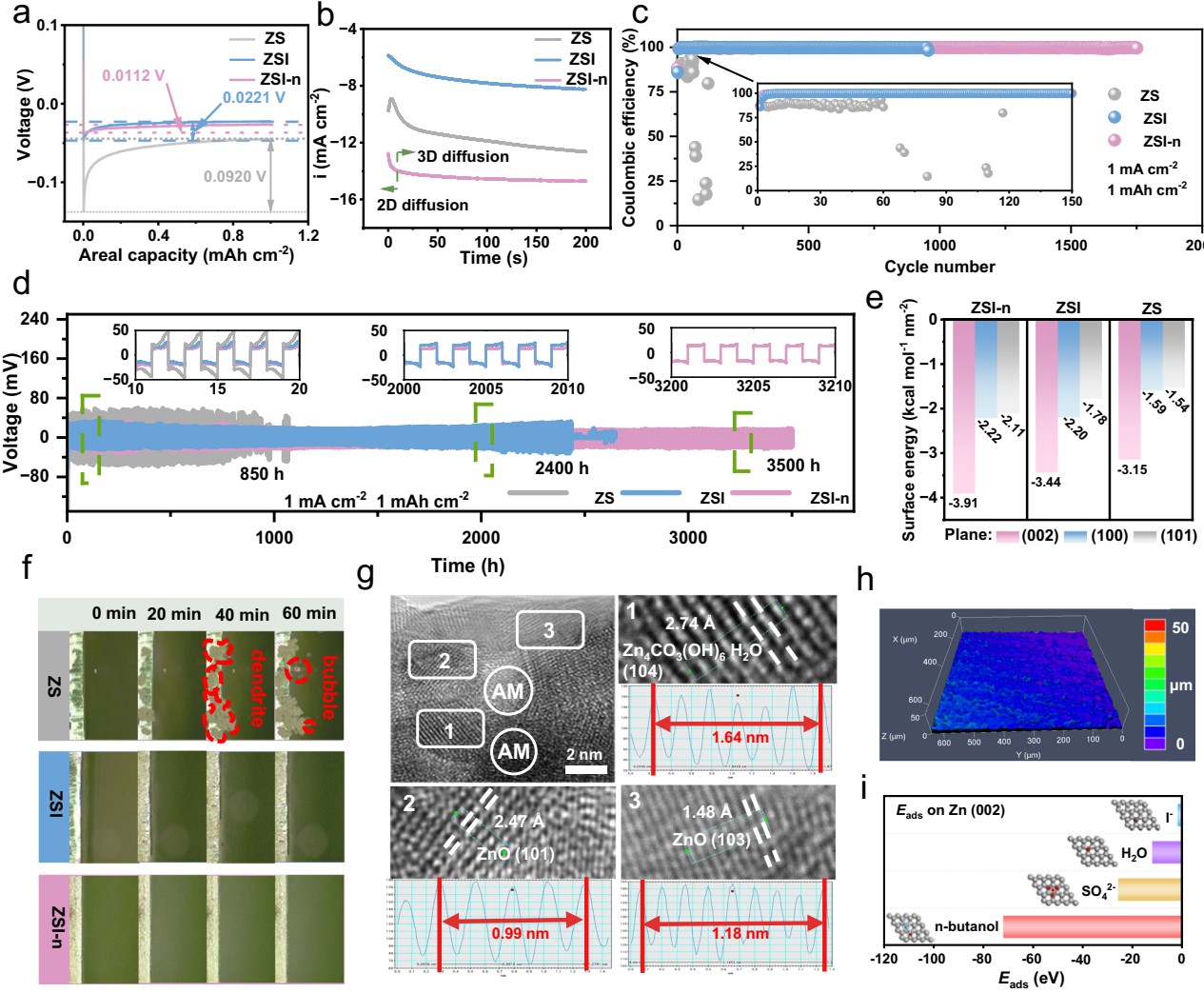

**Fig. 4 | Stability and mechanism analysis of zinc anodes in different electrolytes. a** Voltage profiles of galvanostatic Zn deposition on Cu at 1 mA cm⁻². **b** CA tests in different electrolytes. **c** CE profiles of Zn//Cu cells in three electrolytes. **d** Symmetrical cells operating at f 1 mA cm⁻². **e** Surface energy value for main planes of metallic Zn in three electrolytes. **f** Optical microscopy image of Zn/electrolyte interface during Zn deposition in ZS, ZSI, and ZSI-n. **g** High-resolution TEM images of the Zn anode surface after cycling in three electrolytes. **h** LCSM images of Zn anodes after cycling in ZSI-n. **i** Values of surface adsorption energy of $H_2O$, $SO_4^{2-}$, $I^-$ and n-butanol on Zn (002) plane.

increased number of H-bonds resulting from the introduction of n-butanol[29,32]. Figure 3g, i represents Raman spectra of the three solution systems. The O-H stretching vibrations at 3100–3600 cm⁻¹ exhibit a notable blue shift, indicating that the addition of new molecules disrupts the existing hydrogen bonding network and reduces the activity of water. Simultaneously, the $\nu(SO_4^{2-})$ stretching also shows a slight blue shift, illustrating a tighter binding between $SO_4^{2-}$ and $Zn^{2+}$. Furthermore, the redshift of Zn-OH stretching vibrations (around 400 cm⁻¹) in Fig. 3i further indicates a weakened binding between $Zn^{2+}$ and water in the solvated sheath due to the increase in electron density of water[33], which aligns with the results obtained from NMR and FTIR analyses.

## Stability and mechanism analysis of zinc anodes in different electrolytes

To investigate the influence of electrolytes on zinc anode, Zn//Cu asymmetric cells were employed to elucidate the mechanism of the zinc nucleation stage. As shown in Fig. 4a, the overpotential of ZSI-n reaches an exceptionally low value (11.2 mV), while the overpotential of ZSI is slightly higher (22.1 mV), but still lower to that of ZS (92.0 mV). Besides, a similar decreasing trend of overpotential is observed in CV

tests on Zn//Zn symmetric cells (Supplementary Fig. 24)[34]. A low nucleation overpotential in ZSI-n states a reduction in battery polarization, resulting in uniform zinc deposition in this electrolyte, as scrutinized by chronoamperometry (CA) measurements (Fig. 4b)[35]. The surface-specific current in the ZS electrolyte keeps increasing for 200 s, clarifying that the area of zinc deposition has been expanding, leading to the aggregation of zinc and the formation of dendrites[33]. In contrast, the specific current of the battery using ZSI as the electrolyte significantly slows down over time. The most notable effect is observed when batteries with ZSI-n as the electrolyte tend to stabilize after a rapid increase in specific current within 9 s, attributed to the interaction between n-butanol and the zinc surface inducing 3D deposition of zinc, ultimately forming an even zinc deposition layer[15,36]. Additionally, anti-corrosion capability is essential for the protection and long-term utilization of zinc anode, and the corrosion resistance is obtained from the Tafel curve. According to the fitting results of the polarization curves shown in Supplementary Fig. 25, the battery with ZSI-n has a remarkable effect on the corrosion behavior of zinc anode. Specifically, ZSI-n exhibits a higher corrosion potential (−0.8413 V vs. Ag/AgCl) and a lower corrosion-specific current (1.9445 mA cm⁻²) than those of ZSI (−0.9643 V with 4.6451 mA cm⁻²) and ZS (−0.9658 V with

3.4276 mA cm$^{-2}$), indicating an increased corrosion resistance as well as a good resistance to Zn corrosion of ZSI-n.

The assembled Zn//Cu asymmetric half battery provided insights into the reversibility of the deposition and stripping of Zn on a copper substrate. As depicted in Supplementary Figs. 26 and 4c, the half battery with ZS electrolyte exhibits a substantial overpotential and fails after only 60 cycles due to a low Coulombic efficiency (85%). However, the introduction of I$^-$ and n-butanol reduces the overpotential and enhances the deposition and stripping process, along with an increased Coulombic efficiency (99.99% for ZSI and ZSI-n) due to reduced interface resistance and enhanced ion conductivity, as evaluated in Supplementary Fig. 27[37]. The Zn$^{2+}$ transference number ($t_{Zn^{2+}}$) of ZIS-n shown in Supplementary Fig. 28 is calculated to be 0.38, higher than that of ZSI (0.33) and ZS (0.30)[38]. The increase in the migration number of Zn$^{2+}$ reduces the Zn$^{2+}$ concentration changes during zinc deposition/stripping, thereby making zinc deposition more uniform[39,40]. Moreover, as shown in Supplementary Figs. 29–30, the E$_a$ values for ZSI-n, ZSI, and ZS are 36.01, 41.96, and 56.46 kJ mol$^{-1}$, respectively. The lower E$_a$ indicates an enhanced dynamic process for the electrocatalytic redox reactions of iodine[41,42].

In addition, the Zn//Zn symmetric test served as another way to assess the reversibility of the zinc anode. The cell with ZS electrolyte falls off after continuous high polarization voltage cycling for 850 h at a specific current of 1 mA cm$^{-2}$ with a capacity of 1 mAh cm$^{-2}$ (Fig. 4d). Conversely, upon introducing I$^-$, the polarization voltage of the cell initially steadily decreases and works for 2400 h. Moreover, with the involvement of n-butanol, an ultra-long cycle life of 3500 h with a small polarization voltage is achieved. This indicates the better inhibitory effect on side reactions following the participation of I$^-$ and n-butanol. In addition, the contact angle test was utilized to reflect the zincophilicity of the three electrolytes on the zinc anode (Supplementary Fig. 31). The ZSI-n electrolyte exhibits the lowest contact angle of 69.51° compared to that of ZSI (82.11°) and ZS (100.12°). The better zincophilicity is conducive to rapid electrode infiltration and accelerated ion transport.

During the continuous deposition and stripping process of Zn, its electrochemical characteristics are primarily governed by the surface atomic structure, which is reflected in surface energy[32,43,44]. DFT calculations were performed to determine the surface energy of three main planes of zinc ((002), (100), and (101)) in three electrolyte environments (ZS, ZSI, and ZSI-n) in Fig. 4e. The surface energy of (002) in ZSI-n (Supplementary Fig. 32) reaches a minimum of −3.91 kcal mol$^{-1}$ nm$^{-2}$, in contrast to (100) and (101), which are −2.22 and −2.11 kcal mol$^{-1}$ nm$^{-2}$, respectively. This substantial difference indicates that zinc tends to expose the (002) plane during the growth process to maintain low surface energy. This orientation results in a more even and smooth deposition of Zn$^{2+}$ compared to (100) and (101), showcasing a superior ability to inhibit zinc dendrite formation. Moreover, the same trend of (002) planes being the thermodynamically preferred orientation is observed in the other two electrolyte systems (Supplementary Figs. 33–34). Furthermore, the surface energy of the (002) crystal plane at −3.91 kcal mol$^{-1}$ nm$^{-2}$ with priority growth in the ZSI-n electrolyte is lower than that in ZSI (−3.44 kcal mol$^{-1}$ nm$^{-2}$), and the surface energy of the (002) plane in ZS is the highest (−3.15 kcal mol$^{-1}$ nm$^{-2}$). This explicates that the addition of I$^-$ and n-butanol directs the growth of zinc more toward the (002) plane.

In-situ optical microscopy offered an intuitive perspective to observe the morphological evolution and demonstrate the energetic effects of electrolyte additives, as depicted in Fig. 4f. A significant number of zinc dendrite clusters and bubbles are evident on the surface of the Zn electrode in ZS. In contrast, the zinc electrode deposited in ZSI exhibits a neat surface with few dendrites and no generated bubbles. Furthermore, no excess side reactions occur during zinc

cycling in ZSI-n, resulting in a dense and smooth surface. To further elucidate the morphology of zinc deposition, SEM tests were employed to examine the surface of zinc after plating (Supplementary Fig. 35). The inhomogeneous distribution of large zinc dendrites is formed on the zinc surface in ZS electrolyte, but the zinc surface in ZSI is uniform and even. Furthermore, the inclusion of n-butanol in ZSI results in a tighter and flatter surface, effectively suppressing dendrite growth and extending the lifespan of the battery. The substantial difference in surface characteristics was more pronounced in a laser scanning confocal microscope (LSCM), as shown in Fig. 4h and Supplementary Figs. 36a, b. A pronounced longitudinal drop is observed on the zinc surface in ZS, suggesting the growth of numerous large-sized and uneven dendrites. In contrast, ZSI and ZSI-n exhibit uniformly flat and dense surfaces, with the latter being particularly noteworthy. All these results visually demonstrate the inhibition of zinc dendrites and the promotion of smooth growth of zinc by electrolyte modification.

Then, in order to further explore the working mechanism of electrolyte on the zinc anode, XRD and XPS analyses were conducted on zinc anodes electrodeposited for one hour at 5 mA cm$^{-2}$ in three electrolytes, and compared with original zinc. In Supplementary Fig. 37a, a prominent peak at 8.05° is clearly observed in the curve for ZS, corresponding to the main components of zinc dendrite (Zn$_4$(OH)$_6$SO$_4$·5H$_2$O, JCPDS: 39-0688). However, this peak is absent in the curves after cycling in the other two electrolytes, indicating effective inhibition of zinc dendrite growth by the electrolyte additives. This is further supported by a noticeable decrease in the density of SO$_4^{2-}$ in the S spectrum (Supplementary Fig. 38a). Moreover, three peaks at 36.29°, 38.99°, and 43.22° in Supplementary Fig. 37a correspond to the crystal planes of (002), (100), and (101) (Zn, JCPDS: 99-0110), respectively[45,46]. Notably, the values of the ratio of (002)/(101) after cycling in ZSI-n, ZSI, and ZS (Supplementary Fig. 39) follow a decreasing order (0.37, 0.25, and 0.12, respectively), all higher than that of the original zinc. This further supports the preference of zinc for growing along the (002) crystal plane during the cycling process, aligning with the earlier DFT calculations.

Furthermore, the ZSI-n curve exhibits two new peaks in Supplementary Fig. 37b at 31.73° and 34.37°, corresponding to the presence of zinc oxide (ZnO, JCPDS: 89-1397). Meanwhile, another new peak of ZSI-n at 12.82° indicates the generation of zinc carbonate (Zn$_4$CO$_3$(OH)$_6$·H$_2$O, JCPDS: 11-0287), presenting trace amounts on ZS and ZSI (Supplementary Fig. 37c). The presence of Zn-O bonds in the Zn spectrum (Supplementary Fig. 38b) and the increased ZnCO$_3$ peak in the C spectrum (Supplementary Fig. 38c) of the ZSI-n curve confirm the enrichment of ZnO and Zn$_4$CO$_3$(OH)$_6$·H$_2$O on Zn surface. Additionally, high-resolution transmission electron microscopy (HRTEM) demonstrates that a solid electrolyte interface (SEI) layer is composed of the organic phase and the inorganic phase consisting of ZnO and Zn$_4$CO$_3$(OH)$_6$·H$_2$O, as evidenced by the unique lattice fringe of both observed in the SEI (Fig. 4g). The formation of the organic phase is supported by the strong adsorption capacity of n-butanol on the zinc surface, as revealed by DFT calculations in Fig. 4i. The adsorption energies on Zn (002) for various particles, including n-butanol, SO$_4^{2-}$, H$_2$O, and I$^-$, exhibit significant differences (−72.24, −25.93, −12.21, and −1.99 eV, respectively). The highly negative adsorption energy of n-butanol suggests its easy adsorption on the Zn surface. Furthermore, the photos of three electrolytes are shown in the Supplementary Fig. 40. Where the ZSI-n electrolyte appears opaque, while ZS and ZSI electrolytes appear clear. In addition, optical microscope was used to further characterize the electrolytes. As shown in Supplementary Fig. 41c, there are many small droplets in ZSI-n electrolyte, while ZS (Supplementary Fig. 41a) and ZSI (Supplementary Fig. 41b) electrolytes present a homogeneous solution. These results indicate that ZSI-n is a suspended electrolyte due to the low solubility of n-butanol in aqueous solution. Thus, SEI film

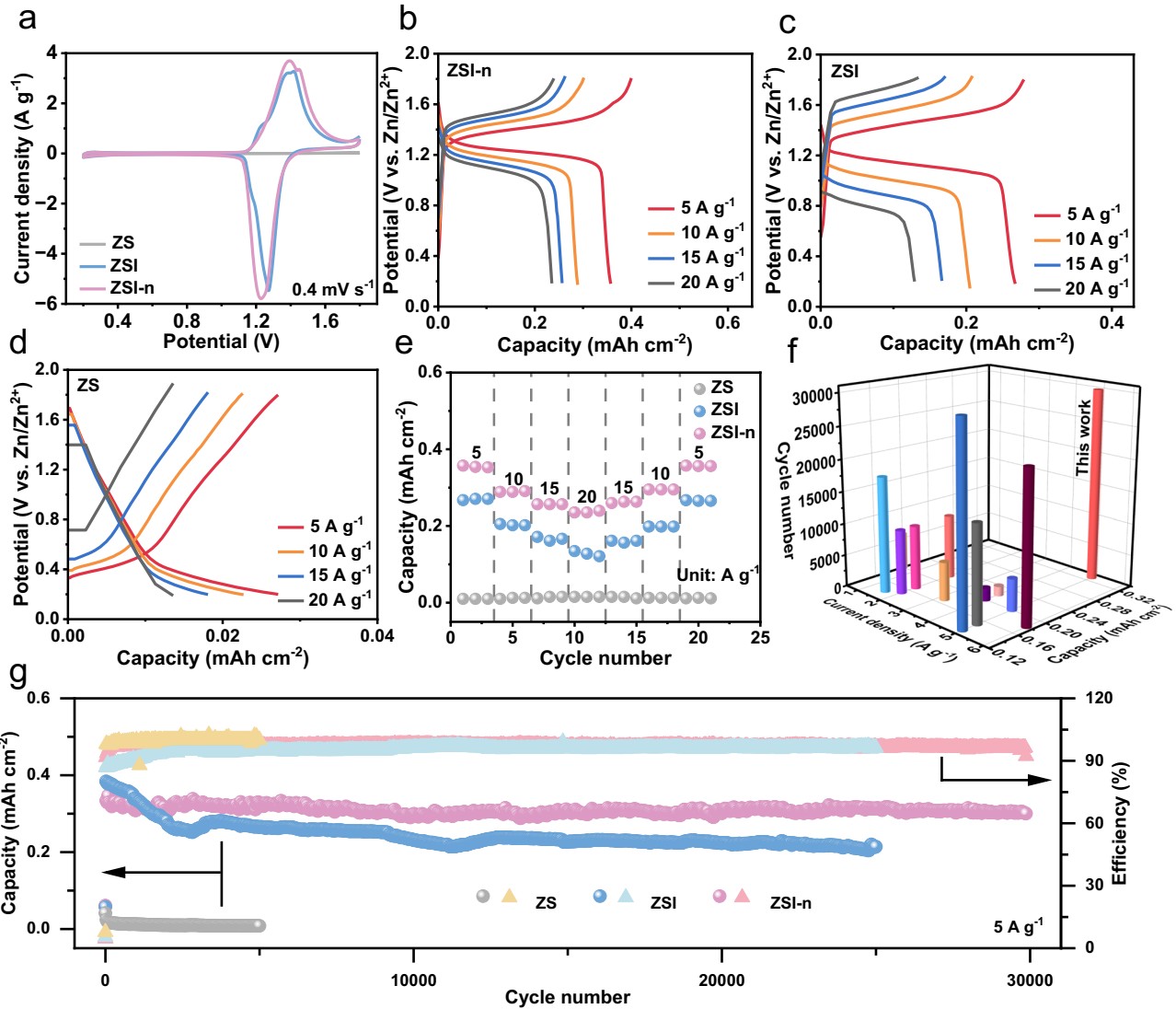

**Fig. 5 | Electrochemical performance of full cell with MXene as cathode host, ZS/ZSI/ZSI-n as electrolyte, and zinc as anode. a** CV curves of ZS, ZSI, and ZSI-n at 0.4 mV s⁻¹. Galvanostatic charge-discharge (GCD) curves of (**b**) ZS, (**c**) ZSI, (**d**) ZSI-n. **e** Rate performance comparison of ZS, ZSI, and ZSI-n. **f** Comparison of cycle number, specific current, and capacity between our battery and other reported Zn-I₂ batteries. **g** cycling capability of full cell with ZS, ZSI, and ZSI-n as electrolytes at 5 A g⁻¹.

containing n-butanol has the robust hydrophobicity, which effectively prevents water molecules from contacting the zinc surface, thereby suppressing unpleasant side reactions[47]. Overall, inorganic ZnO, $Zn_4(OH)_6SO_4 \cdot 5H_2O$, and organic additives n-butanol successfully form an effective SEI layer, further inducing uniform deposition of zinc towards (002) crystal plane and effectively suppressing the accompanying growth of zinc dendrites.

## Assembly and performance study of Zn-I₂ full cells

The preceding discussion has underscored the superiority of TMX cathode host, electrolyte modification, and anode protection. To substantiate the synergistic impact of the three factors when used in tandem in the battery, we employed TMX as the cathode host, zinc as the anode electrode, and incorporated three different electrolytes for full cell testing. The cyclic voltammetry (CV) curve of the battery with ZS as the electrolyte reveals almost no discernible capacity, indicating that $Zn^{2+}$ does not contribute to the capacity of this battery. In contrast, a pair of redox peaks at 1.27/1.41 V and 1.22/1.39 V is evident in the curve with ZSI and ZSI-n as electrolytes, respectively. Moreover, the

larger integration area of the curve with ZSI-n indicates a higher capacity it possesses (Fig. 5a). This may be due to the suspended n-butanol restricts $I_3^-$ near the TMX cathode host, resulting in no significant change in the concentration of $I_3^-$ near the cathode region compared to ZSI without the restriction of suspended n-butanol. This suppresses the conversion of $I_2$ to $I_3^-$ on the TMX cathode host, which reduces the loss of iodine load on TMX and improve the capacity performance.

It should be noted that although there is no $I_2$ added in the initial full cells, the $I^-$ in the electrolyte can be converted to $I_2$ deposited on the MXene host during charging process. The in-situ electrodeposition enables $I_2$ to grow uniformly on the surface within MXene, enhancing its conductivity and ion diffusion ability[8]. More importantly, in fully charged state, the mass loading of $I_2$ is as high as 1.45 mg cm⁻² (calculation methods and data in Note 6, Supplementary Fig. 42), which is higher than most reported similar batteries (as shown in Table S2). As depicted in Fig. 5b, the battery in ZSI-n achieves a higher Coulombic efficiency of 95.12% with a larger charge capacity (0.41 mAh cm⁻²) and discharge capacity (0.39 mAh cm⁻²) than that of ZSI (0.28 mAh cm⁻²

and 0.26 mAh cm$^{-2}$ shown in Fig. 5c), proving the benefits of n-butanol for improving performance. However, the capacity of ZS in Fig. 5d is one order of magnitude lower than the other two. Impressively, the battery with ZSI-n delivers a high discharge capacity of 0.35, 0.29, 0.26, and 0.24 mAh cm$^{-2}$ at 5, 10, 15, and 20 A g$^{-1}$, respectively, while the slightly lower discharge capacity for ZSI is 0.26, 0.20, 0.17, and 0.12 mAh cm$^{-2}$ at the corresponding current densities (Fig. 5e). The capacity improvement brought by the introduction of n-butanol electrolyte additives can be mainly attributed to the inhibition of the shuttle effect of I$_3^-$. In addition, it should be noted that the mass loading of I$_2$ on the cathode can be adjusted by changing the electrolyte concentration in this study. As shown in Supplementary Fig. 43, the mass loading of I$_2$ reaches 38.5 mg cm$^{-2}$, which is the highest value reported by far[48]. More importantly, while with such a high mass loading, the battery exhibits a specific capacity of 125.9 mAh g$^{-1}$.

The electrochemical impedance spectroscopy (EIS) curves of the three cells are also presented in Supplementary Fig. 44, where the lowest values of R$_s$ and R$_{ct}$ indicate that ZSI-n owns the fastest transfer of electrons, followed by ZSI, and the slowest in ZS. Long-term cycle tests (Fig. 5g) show the ultra-long lifespan of ZSI-n for 30,000 cycles with a high capacity of 0.30 mAh cm$^{-2}$ and a small capacity decay of 0.0004% per cycle. Additionally, the performance in ZSI without n-butanol works for 25,000 cycles with a higher decay (0.003% per cycle) and a slightly inferior capacity of 0.22 mAh cm$^{-2}$. Our study is compared with previous reports, as illustrated in Fig. 5f. The capacity under high specific current surpasses that of most reported results, even those obtained under low specific current[6–8,23,28,38,42,49–53]. The highly advanced full battery performance confirms the success of our tripartite synergistic optimization strategy on Zn-I$_2$ battery.

## Discussion

We have developed a tripartite synergistic optimization strategy involving cathode host, electrolyte additive, and in-situ anode protection to solve the problem of short cycle life of Zn-I$_2$ batteries at high specific current. TMX, serving as a cathode host without any active materials, offers ample active sites for iodine generation, improves reaction kinetics, and restricts the movement of the by-product I$_3^-$ toward anode electrode. Meanwhile, the suspended n-butanol further synergizes with MXene to limit the movement of I$_3^-$ due to its strong adsorption capacity for I$_3^-$. Furthermore, the n-butanol dissolved in electrolyte, with its electron-rich groups, facilitates the desolvation process by reducing the binding between Zn$^{2+}$ and water in the solvation structure. Simultaneously, on the zinc anode surface, the adsorption of n-butanol induces a chance in surface energy of different crystal planes, promoting the growth of zinc towards the (002) crystal plane. And a protective solid electrolyte interface (SEI) is formed on the zinc surface, effectively inhibiting the formation of zinc dendrites. Overall, we have profoundly revealed the success of synergistic effects in constructing advanced Zn-I$_2$ batteries through a combination of theoretical calculations and practical applications. Consequently, the full battery demonstrates remarkable performance, achieving a capacity of 0.30 mAh cm$^{-2}$ and an energy density of 0.34 mWh cm$^{-2}$ over 30,000 cycles with a low capacity decay of 0.0004% per cycle. Our work provides a promising synergistic approach that opens up new possibilities for high-capacity and long-life Zn-I$_2$ batteries operating at high current densities.

## Methods

### Materials

Titanium aluminum carbide powder (Ti$_3$AlC$_2$, 11 Technology Co., Ltd.), lithium fluoride (LiF, Aladdin), hydrochloric acid (HCl, Chengdu Chron Chemical Co., Ltd.), zinc sulfate (ZnSO$_4$, Aladdin), zinc iodide (ZnI$_2$, Maclin), n-butanol (Aladdin), N-methyl-2-pyrrolidone (NMP, Aladdin) and poly-(vinylidene fluoride) (PVDF, Aladdin) are used directly from the opening without further treatment.

### Preparation of Ti$_3$C$_2$T$_x$ MXene

1 g LiF was added in 40 mL HCl solution, followed by the addition of 1 g Ti$_3$AlC$_2$ slowly. The mixture was maintained at 40 °C for 40 h in the oil bath and cooled down naturally. The obtained suspension was washed with deionized water at 3500 rpm for 5 min until the pH≈6. And the sediment was ultrasonic for 20 min at 300 W to further delaminate. After another centrifugation, the Ti$_3$C$_2$T$_x$ MXene suspension was obtained.

### Preparation of Ti$_3$C$_2$T$_x$ MXene electrode

The MXene suspension was dried at 60 °C for 12 h to obtain MXene power. The electrode was prepared in the mortar by mixing MXene power, Ketjen black, and PVDF in a mass ratio of 7:2:1 with the adding of NMP in the air atmosphere, and then cost the slurry on carbon cloth. After drying at 60 °C for 12 h in a vacuum oven, the carbon cloth covered by mixture was cut into a disc (φ = 12 mm) to be used as the cathode electrode by automatic coating machine. The mass loading of MXene on each disc was about 1 mg/cm$^2$.

### Preparation of electrolytes

Zinc sulfate was dissolved into the deionized water to obtain the 2 M ZnSO$_4$ named ZS. 0.2 M ZnI$_2$ was added into the previous ZS electrolyte to get ZSI electrolyte. 3% (volume) n-butanol was introduced to ZSI electrolyte to obtain the electrolyte called ZSI-n electrolyte. Electrolyte is transferred using a pipette in an air atmosphere at 25 °C and prepared using an oscillator. The prepared electrolyte is stored in a sealed brown bottle for preservation in an air atmosphere at 25 °C.

### Materials characterization

UV-Vis was recorded on a PE lambda 750 UV-Vis spectrophotometer. Fourier transform infrared (FTIR) mapping was performed on Thermo Scientific Nicolet iS20 infrared spectroscopy instrument with the KBr pellet technique. Raman spectroscopy was collected on Horiba Lab-RAM HR Evolution with a laser wavelength of 532 nm. $^1$H Nuclear magnetic resonance spectrum was carried out on a Bruker AVANCE 400 spectrometer using deuterated DMSO as the field frequency lock (400 MHz).

### Electrochemical measurements

The half-cell, Zn//Zn cell, and Zn//Cu cell were all assembled in coin cell (CR2032). Commercial zinc foil can be directly used after being cut by a cutting machine. Galvanostatic charge/discharge curves (GCD), rate performance, and long-term cycling tests were performed at NEWARE battery tests. Cyclic voltammograms (CV), electrochemical impedance spectra (EIS), and chronoamperometry (CA) were investigated on an electrochemical workstation (CHI760). The upper limit of voltage is 1.8 V, and the lower limit of voltage is 0.2 V. The specific current is 5 A g$^{-1}$. The electrochemical energy storage tests are carried out in an environmental chamber and the cell cycling experiments are carried out at 25 °C. The thickness, diameter, area, mass loading, and active material of cathode electrode are 0.4 mm, 12 mm, 1.13 cm$^{-2}$, 1.45 mg cm$^{-2}$, and 70 wt% respectively. The thickness and area of the anode zinc electrode are 0.03 mm and 1.13 cm$^{-2}$ respectively. The thickness, diameter, and area of the glass-fiber separator is 1 mm, 16 mm, and 2.01 cm$^{-2}$, and the amount of electrolyte droplets added to each coin cell is 80 μL.

### DFT calculation for adsorption energy and charge density difference

The I$_3^-$ adsorption calculations on single-layer graphene and -OH functional Ti$_3$C$_2$ surface were respectively performed within the framework of density functional theory (DFT) as implemented in the Vienna Ab-initio Simulation Package (VASP) code by using the projector augmented wave method with the Perdew-Burke-Ernzerhof (PBE) exchange-correlation functional[54–56]. The influence of vdW

interactions is considered by using a modified version of vdW-DF, referred to as "optB86b-vdW" [57,58]. The projector augmented wave potentials[59] were used with an energy cutoff of 600 eV. A $4 \times 4 \times 1$ Monkhorst-Pack method k-mesh was used for geometry optimization of single-layer graphene and -OH functional $Ti_3C_2$. Energy convergence of $1.0 \times 10^{-4}$ meV/atom was ensured during the self-consistent field calculations. And the convergence criteria for the atomic forces was 0.01 eV/Å. Then one $I_3$ molecule was respectively adsorbed on graphene and $Ti_3C_2$ slab surface. There exists a vacuum layer of large than 20 Å perpendicular to the surface plane. Geometry optimization was then performed for each adsorption system followed by static calculations. A $8 \times 8 \times 1$ Monkhorst-Pack method k-mesh was used for single-point energy. The adsorption energies $E_{ad}$ were calculated according to the following equation[60,61]:

$$E_{ad} = E_{total} - E_{molecule} - E_{slab} \tag{1}$$

Where $E_{total}$, is the total energy of the adsorption system, $E_{molecule}$ and $E_{slab}$ are the energies of adsorbed adsorbate, i.e., $I_3$ and $Ti_3C_2$ slab, respectively. The charge density difference $\Delta\rho$ was calculated according to the following equation:

$$\Delta\rho = \rho_{total} - \rho_{molecule} - \rho_{slab} \tag{2}$$

Where $\rho_{total}$, is the total charge density of the adsorption system, $\rho_{molecule}$ and $\rho_{slab}$ are the charge densities of adsorbed adsorbate, i.e., $I_3$ and $Ti_3C_2$ slab, respectively. The charge density differences were visualized by VESTA with an isosurface value of 0.0075 e/Å³.

### Molecular dynamic (MD) simulations for solvation structure analysis

Molecular dynamic (MD) simulations were respectively applied to investigate the solvation structures for three considered electrolyte systems, i.e., S1, S2, and S3. The three electrolyte systems consisting of different numbers of compositions (as shown in Table S3) were respectively constructed into cubic simulation boxes. All MD simulations were carried out by Forcite module with COMPASS III force field[62,63] in MS 2020. Van der Waals and Coulomb interactions were respectively considered by atom-based and Ewald methods with a cut-off value of 12.5 Å. Equations of motion were integrated with a time step of 1 fs. After energy minimization, each system was fully relaxed under periodic boundary conditions for 400 ps in the NPT ($P = 1$ atmosphere, $T = 298.0$ K) ensemble using the Nose thermostat and Berendsen barostat, which was long enough for system temperature, potential and total energy to get stable. After reaching equilibrium state, another 400 ps simulation under NVT ensemble was performed to extract trajectory and data radical distribution function (RDF) and coordination number (CN) calculation. The dynamic trajectory for each system was outputted at an interval of 4 ps. The coordination number $N_i$ of molecules i in the first solvation shell surrounding $Zn^{2+}$ was calculated as:

$$N_i = 4\pi\rho \int_0^{R_M} g(r)r^2 dr \tag{3}$$

in which $R_M$ is the distance of the first minimum following the first peak in the RDF $g(r)$ and $\rho$ is the number density of molecules i[64]. The representative solvation structures were derived from the stable electrolyte configurations.

### MD calculation for three considered electrolytes on Zn (002), (100), and (101) surface

The solid-liquid interface calculations for three considered electrolytes on Zn (002), (100), and (101) surfaces were respectively conducted with COMPASS II force field[58,62] using the Forcite tools in MS

2020[65]. The dimensional lengths of the Zn (002), (100) and (101) slab were $36.00 \text{ Å} \times 35.62 \text{ Å}$, $35.63 \text{ Å} \times 37.78 \text{ Å}$ and $34.46 \text{ Å} \times 40.09 \text{ Å}$ in plane, respectively. After geometry optimization, the three considered electrolytes containing the same number of components as Part One were respectively placed on the Zn (002), (100), and (101) surfaces. All MD calculations were performed under the NVT ensemble ($T = 298.0$ K) with a time step of 1 fs and a total simulation time of 300 ps, during which simulation trajectories were recorded every 3000 steps. The running time was long enough for system energy and temperature reaching stable. The temperature was controlling by a Nose-Hoover thermostat. The Ewald scheme and atom-based cutoff method (i.e., a radius of 12.5 Å) were applied to treat electrostatic and van der Waals (vdW) interactions, respectively. The surface energy $\gamma_{sol}$ for electrolytes on Zn (002), (100), and (101) surfaces was defined by:

$$\gamma_{sol} = \gamma_s + \frac{Esol}{A} \tag{4}$$

Where $\gamma_s$ is the surface energy for Zn (002), (100), and (101) surfaces given by

$$\gamma_s = \frac{E_s^{unrelax} - NE_b}{2A} + \frac{E_s^{relax} - E_s^{unrelax}}{2A} \tag{5}$$

And $E_{sol}$ is calculated by

$$E_{sol} = E_{slab/sol} - E_{slab} - E_{sol} \tag{6}$$

where A is the area of the surface considered, $E_s^{relax}$ and $E_s^{unrelax}$ are energies for, respectively, relaxed and unrelaxed surfaces, N is the number of atoms in the slab and $E_b$ is the bulk energy per atom, $E_{slab/sol}$ is energy of the surface covered with electrolyte, $E_{slab}$ energy for the Zn (002), (100) or (101) clean surface, and $E_{sol}$ energy for the electrolyte.

### Reporting summary

Further information on research design is available in the Nature Portfolio Reporting Summary linked to this article.

## Data availability

All data that support the findings of this study are provided within the paper and its Supplementary Information. All additional information is available from the corresponding authors upon request. Source data are provided with this paper.

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

## Acknowledgements

This work was supported by the National Natural Science Foundation of China (62201369, 52072252), the Department of Science and Technology of Sichuan Province (2024NSFSC0226, 2023NSFSC1942), and the Fundamental Research Funds for the Central Universities (YJ2021100).

## Author contributions

Z.L., J.Z., and W.Y. conceived the idea. Y.L. supervised the research. W.Y., J.Q., and F.T. conducted the synthesis and characterization. W.Y., F.T., and X.C. performed the electrochemical performance tests. J.Q. and J.L. carried out the computations. W.Y. organized the figures. W.Y. and J.Z. wrote the manuscript. C.D., Z.L., and J.Z. revised the manuscript. All authors contributed to the manuscript preparation.

## Competing interests

The authors declare no competing interests.
