## [Transparent Peer Review file · Nature Communications]

A tripartite synergistic optimization strategy for zinc-iodine batteries

Corresponding Author: Professor Zifeng Lin

Version 0:

Reviewer comments:

Reviewer #1

(Remarks to the Author)

This manuscript employs a tripartite synergistic optimization strategy to comprehensively improve the cycling stability of Zn-I₂ batteries. This is achieved by adjusting the solvation structure and preventing Zn metal dendritic growth using MXene and n-butanol. These strategies are commonly used in the literature with various additives and have been proven effective. A critical issue for this manuscript is the estimated energy density of the designed Zn-I₂ battery. The manuscript claims a capacity of 0.30 mAh cm⁻² at 5 A g⁻¹ with an excellent capacity retention of 88.23%. Additionally, only 0.02 M ZnI₂ was added to the existing ZS electrolyte to create the ZSI electrolyte. Therefore, it is important to determine the mass loading of I₂ participating in the reaction, as discussing small amounts of I₂ participation is not meaningful. Hence it's not recommended to be published in a high quality journal of Nature Communication.

1. The title of the manuscript and the supporting information do not match. Please verify and ensure consistency across both documents.
2. The manuscript extensively discusses the impact of the electrolyte composition, specifically the addition of ZnI₂ and n-butanol, on the zinc anode. However, it lacks an explanation of how electrolyte optimization could address critical issues in Zn-I₂ batteries, such as the shuttling of polyiodide ions.
3. The experimental section notes that the cathode does not contain I₂, and the only source of iodide (I⁻) is from the ZnI₂ additive. Could you specify the active material mass in the battery system? Given that ZnI₂ is used in a small amount, how does this affect the energy density of the battery? An analysis or discussion on this aspect would enhance the understanding of the performance of the battery.
4. The manuscript highlights the issue of I₃⁻ ion shuttling in Zn-I₂ batteries, yet it provides insufficient details on how I₃⁻ shuttles between electrodes. A more thorough explanation of this shuttle mechanism is necessary to contextualize the discussion properly.
5. The manuscript asserts that the TMX cathode host, characterized by its layered structure and surface-rich groups, facilitates the I⁻/I₂ conversion reaction, which purportedly reduces I₃⁻ formation. However, this claim lacks empirical backing. It is advisable to include in situ or ex situ Raman spectroscopy, XRD, or XPS data to provide evidence of the iodine conversion mechanism.
6. In Figure 5f, the authors present a comparison of cycle number, current density, and capacity among various reported Zn-I₂ batteries. However, specific references for the comparative data are missing. Please provide these references and the comparative data for further reading. It is noted that the paper uses additives as active materials, which qualitatively differ from the materials typically used in iodine cathodes. Since iodine cathodes generally contain a higher mass of active substances, comparing these different systems may not provide a meaningful analysis. Please reconsider the basis for this comparison or provide a detailed rationale for including such disparate systems in the comparative analysis to ensure the validity and relevance of the discussion.

Reviewer #2

(Remarks to the Author)

The manuscript presents a tripartite synergistic optimization strategy involving a cathode host, electrolyte additive, and in-situ formed zinc anode protective layer. This approach addresses challenges related to iodine dissolution and zinc dendrite growth in zinc-iodine batteries, resulting in exceptional electrochemical performance characterized by high capacity and extended cycle life. The authors also elucidate the charge storage mechanisms using those techniques. Overall, this paper holds promise for publication in Nature Communications following some revisions. Below are the detailed comments for the authors:

1. The area capacity in ZSI electrolyte is reported as 0.22 mAh cm⁻², while introducing the n-butanol additive increases it to

0.30 mAh cm⁻². Please explain the rationale behind this enhancement in area capacity upon adding the electrolyte additive.

2. The authors determine the thickness of TMX to be 1.9 nm using AFM. How does the thickness impact electrochemical behaviors, and is a thinner TMX preferable for better electrochemical performance?

3. More experimental details are necessary for result reproducibility. For instance, the diameter and thickness of the zinc electrode, electrolyte volume, and separator diameter.

4. The captions corresponding to the three colors in figures 4d and 5e appear to be inaccurate. While an increase in overpotential is expected in symmetric battery testing due to residual SEI accumulation and electrolyte depletion, explain the observed trend of decreasing and then increasing polarization voltage corresponding to the blue and pink colors.

5. When discussing the adsorption of TMX to I³⁻ using UV spectroscopy, the authors should cite relevant literature, particularly regarding the peaks observed in the UV spectra.

6. Please provide corresponding equivalent circuits to fit the Nyquist diagram in the supplementary information section.

7. Please explain the potential advantages of suspended n-butanol over fully dissolved one. Elucidate the underlying principles supporting this distinction.

8. Several errors should be corrected in the text, such as "Zinc-iodine" in the abstract section, "separator" in Figure 1. Ensure references include complete information, including year, volume, and page numbers.

Reviewer #3

(Remarks to the Author)

This study proposes a new strategy that comprehensively improves the performance of zinc-iodine batteries from multiple aspects, involving a MXene cathode host, a n-butanol electrolyte additive and the in-situ solid electrolyte interface. The prepared battery exhibits enhanced capacity as well as long lifespan, and the corresponding mechanism has also been studied and calculated. The idea is interesting and of interest for the readers of the journal. Accordingly, this work could be accepted for publication provided the following issues are solved:

1. The capacity of the battery reaches 0.3 mAh cm⁻² at a high current density of 5 A g⁻¹, which looks good. But what I'm curious about is, will the capacity be larger at lower current densities? For example 1 A g⁻¹.

2. More specific measurements should be used to study the protection of the anode electrode by the in-situ generated SEI film, such as its corrosion resistance.

3. The author claims that the MXene host and n-butanol additive inhibit the shuttle of I³⁻ synergistically, so which one contributes more?

4. There have been several reports on alcohol electrolyte additives before (for example, Energy Storage Mater. 48 (2022) 192-204; Angew. Chem. Int. Ed. 2021, 60, 7366-7375; Adv. Funct. Mater. 2023, 2214538). What is the advantage of n-butanol additives compared to previous reports?

5. Authors should explain why 3% v/v n-butanol is used in the electrolyte. Have experiments been done with other percentages of n-butanol? How does the amount of n-butanol affect the battery?

6. The author mentioned in the manuscript that ZSI-n is a suspended electrolyte. If the author provides additional optical photos of three electrolytes, it will be beneficial for readers to understand.

7. The author should elaborate on the significance of the increased Zn²⁺ transference number during the zinc deposition/stripping process, which will provide crucial insights into the mechanism behind the battery's improved performance and long cycle life.

8. The author analyzed the changes in solvation structure through theoretical calculations and structural characterization, but did not explain the effects of solvation structure. It is better to add the corresponding discussion in the "Characterization and Simulation of Electricity Structure" section.

9. The performance of the prepared Zn-I₂ battery appears to be good, but there is a lack of comparative data with similar types of batteries. It is recommended to add a performance comparison table with other latest reports.

Version 1:

Reviewer comments:

Reviewer #1

(Remarks to the Author)

I sincerely appreciate the authors' thorough explanations and the additional data provided. However, I must express disappointment regarding the error in the ZnI₂ concentration, as this is a critical factor for the experiments and is important for readers to assess the study accurately.

Furthermore, the authors state, "the mass loading of I₂ and the areal energy density of the battery are as high as 1.45 mg cm⁻² and 0.34 mWh cm⁻², respectively, which are higher than most current Zn-I₂ batteries." However, a high-loading cathode (37.5 mg iodine cm⁻²) has already been achieved and published in Advanced Materials (2024, 36, 2404011). It is

not reasonable to claim that this lower performance merits publication in the high-quality journal Nature Communications

Reviewer #2

(Remarks to the Author)

All concerns have been addressed effectively, and I advocate for the publication of this manuscript.

Reviewer #3

(Remarks to the Author)

The manuscript has been improved, and it can be accepted now.

Point-by-Point Response to Referee Reports

Reviewer #1 (Remarks to the Author):

This manuscript employs a tripartite synergistic optimization strategy to comprehensively improve the cycling stability of Zn-I₂ batteries. This is achieved by adjusting the solvation structure and preventing Zn metal dendritic growth using MXene and n-butanol. These strategies are commonly used in the literature with various additives and have been proven effective.

A critical issue for this manuscript is the estimated energy density of the designed Zn-I₂ battery. The manuscript claims a capacity of 0.30 mAh cm⁻² at 5 A g⁻¹ with an excellent capacity retention of 88.23%. Additionally, only 0.02 M ZnI₂ was added to the existing ZS electrolyte to create the ZSI electrolyte. Therefore, it is important to determine the mass loading of I₂ participating in the reaction, as discussing small amounts of I₂ participation is not meaningful. Hence it's not recommended to be published in a high quality journal of Nature Communication.

Response: We appreciate the critical comments. Although MXene actives or alcohol additives have been reported for the optimization of zinc batteries, the majority of these reports primarily concentrates on enhancing either the cathodes, anodes, or electrolytes, often overlooking the importance of addressing these issues from multiple perspectives synergistically. Furthermore, as the reviewer mentioned, various alcohol additives have been reported in other literatures to adjust the dissolved structure of the electrolyte, and they showed effectiveness in zinc anode protection. However, since these alcohol additives are easily soluble in aqueous solutions, they are difficult to play an effective role in inhibiting the shuttle of I₃⁻ ions, which is also one of the key problems for Zn-I₂ batteries.

Unlike previous reports, we propose a tripartite synergistic optimization strategy to comprehensively improve the performance of Zn-I₂ batteries, which involves the optimization of cathode host, electrolyte, and *in-situ* SEI protection. On the cathode

side, *n*-butanol and MXene synergistically inhibit the shuttle of I_3^- . In the electrolyte, *n*-butanol and I^- synergistically regulate the solvation structure. On the anode side, adsorbed *n*-butanol on the zinc surface and *in-situ* SEI synergistically protect the zinc anode. Through this tripartite synergistic optimization strategy, the constructed Zn-I₂ battery can achieve a high areal capacity of 0.30 mA cm⁻² after 30,000 cycles, surpassing most reported similar batteries (as shown in Table 1). Furthermore, we have updated the introduction to clearly distinguish our work from existing literature.

For another reviewer concern, the mass loading of I₂ and the energy density of battery, we have calculated these important properties and compared with other reported Zn-I₂ batteries in recent years. Although there is no I₂ added in the initial full cell, the I^- in the electrolyte can be converted to I₂ deposited on the MXene host during charging process. As shown in Table 1, the mass loading of I₂ and areal energy density of the battery are as high as 1.45 mg cm⁻² and 0.34 mWh cm⁻², respectively, which are higher than most current Zn-I₂ batteries.

Here we use the MXene host without active I₂ loading due to its unique advantages. Reported methods for I₂ loading generally fall into two categories: (1) mixing I₂ with conductive agents and adhesives and coating them onto a substrate, and (2) placing the cathode electrode host in an iodine vapor atmosphere to load I₂ through the host's adsorption capacity. The main disadvantage of the first method is significant I₂ loss during drying and insufficient adsorption by the adhesive, leading to rapid capacity decay during cycling. Although the second method can address weak adsorption capacity, relying solely on the host's adsorption can result in uneven iodine distribution and damage to the channels (*Adv. Mater.* **2021**, *33*, 2006897). Compared with these two methods, the *in-situ* electrodeposition in this study enables iodine to grow uniformly on the surface and within MXene, enhancing its conductivity and ion diffusion ability. More importantly, the amount of I₂ on the cathode prepared by the *in-situ* electrodeposition method can be flexibly adjusted according to the concentration of I^- in the electrolyte. In addition, we apologize for the mistake that the concentration of ZnI₂ was written incorrectly in the original manuscript, which caused significant confusion for the reviewer. Upon reviewing the comments, we realized the correct

concentration of ZnI_2 should be 0.2 M, instead of 0.02 M. We have corrected this in the manuscript and verified the accuracy of other experimental parameters. The experimental process for the 0.2 M ZnI_2 concentration has been added to the manuscript (Note 2 and Supplementary Figs. 8-9).

Changes:

While the current reports have made some progress in improving performance of Zn-I₂ batteries, the majority of current research primarily concentrates on enhancing either the cathode electrode, anode electrode, or electrolyte, often overlooking the importance of addressing the other part. (Pages 5 of the revised Manuscript)

Consequently, the fabricated full batteries exhibit an outstanding cycle life of 30,000 cycles with a high capacity of 0.30 mAh cm⁻² and a superior energy density of 0.34 mWh cm⁻² at 5 A g⁻¹ with a superb capacity retention of 88.23%, surpassing other reported Zn-I₂ batteries. (Page 6 of the revised Manuscript)

It should be noted that the concentration of ZnI_2 (0.2 M) was selected through experimental exploration, and the corresponding discussion and data are presented in Note 2 and Supplementary Figs. 8-9. (Page 10 of the revised Manuscript)

It should be noted that although there is no I₂ added in the initial full cells, the I⁻ in the electrolyte can be converted to I₂ deposited on the MXene host during charging process. The *in-situ* electrodeposition enables I₂ to grow uniformly on the surface within MXene, enhancing its conductivity and ion diffusion ability⁸. More importantly, in fully charge state, the mass loading of I₂ is as high as 1.45 mg cm⁻² (calculation methods and data in Note 6, Supplementary Fig. 42), which is higher than most reported similar batteries (Table 1). (Page 25 of the revised Manuscript)

Table 1. Summary of the optimization strategy, load mass, current density, area capacity density, area energy density and cycle number of Zn-I₂ batteries reported recently. (Pages 45 of the revised SI)

Reference	Optimization strategy	Load mass (mg cm ⁻²)	Current density (A g ⁻¹)	Area capacity density (mAh cm ⁻²)	Area energy density (mWh cm ⁻²)	Cycle number
[1]	Cathode	0.06	1	0.014	0.016	1500
[2]	Cathode Anode	1.0	2.5	0.118	0.13	3000
[3]	Cathode	1.0	2.11	0.175	0.21	10000
[4]	Anode	0.63	2	0.126	0.15	10000
[5]	Electrolyte additive In-situ SEI	1.38	1.45	0.259	0.30	5000
[6]	Cathode	1.0	5	0.095	0.11	20000
[7]	Anode	1.3	2	0.135	0.15	20000
[8]	Anode	1.0-2.0	3.2	0.09-0.17	0.10-0.20	6000
[9]	Cathode	1.0	1	0.2	0.25	5700
[10]	Cathode	1.0	4.22	0.14	0.16	10000
[11]	Cathode	0.8-1.5	2	0.16-0.3	0.19-0.35	10000
[12]	Cathode	1.25	4	0.25	0.28	1500
[13]	Gel electrolyte	1.5	1.58	0.24	0.26	5000
[14]	Cathode	2	5	0.24	0.29	5000
[15]	Cathode	2	8.44	0.16	0.19	6000
[16]	Cathode	1.0-1.2	2.11	0.15-0.18	0.16-0.20	20000
[17]	Cathode	1.17	4.22	0.154	0.20	5000
[18]	Cathode	1.0-1.2	1.2	0.21-0.25	0.27-0.32	1000
[19]	Cathode	1.0-1.2	5	0.16-0.19	0.19-0.22	30000
[20]	Gel electrolyte	0.8-1.0	2.11	0.12-0.15	0.14-0.17	22000
[21]	Ionic liquid electrolyte	0.8-1.0	4	0.12-0.15	0.14-0.17	18000
[22]	Cathode	1.0-1.2	5	0.17-0.2	0.20-0.24	17000
[23]	Cathode	1.0-2.0	2	0.1-0.2	0.12-0.23	1400
[24]	Cathode	0.8-1.2	3	0.1-0.15	0.12-0.18	5900
This work	Cathode Electrolyte additive In-situ SEI	1.45	5	0.30	0.34	30000

The corresponding references are listed in the supporting information.

Supplementary Fig. 8 Capacity and Coulombic efficiency of different contents of ZnI₂ (2 M ZnSO₄+0.2 M ZnI₂, 2 M ZnSO₄+0.5 M ZnI₂ and 2 M ZnSO₄+1.0 M ZnI₂) (Page 9 of the revised SI)

Supplementary Fig. 9 Zn//Zn symmetrical tests of different contents of ZnI₂ (2 M ZnSO₄+0.2 M ZnI₂, 2 M ZnSO₄+0.5 M ZnI₂ and 2 M ZnSO₄+1.0 M ZnI₂) (Page 10 of the revised SI)

Note 2: Three electrolytes with different contents of ZnI₂ (2 M ZnSO₄+0.2 M ZnI₂, 2 M ZnSO₄+0.5 M ZnI₂ and 2 M ZnSO₄+1.0 M ZnI₂) were prepared for long cycle testing and Zn//Zn symmetric testing. From the Supplementary Fig. 8, it can be seen that electrolytes containing different concentrations of ZnI₂ have little effect on capacity, and even the capacity decreases in electrolytes containing high concentrations of ZnI₂. Besides, the battery with an electrolyte containing 0.2 M ZnI₂ exhibits a high Coulombic efficiency of 93%, which is much higher than the other two electrolytes (0.5 M ZnI₂: 12%; 1.0 M ZnI₂: 2%). This is because high concentrations of I⁻ are more likely

to combine with I_2 to form I_3^- , leading to overcharging and the production of more I_3^- , resulting in severe shuttle effects. In addition, the effect of different concentrations of ZnI_2 on the zinc anode is further investigated. As shown in the Supplementary Fig. 9, symmetric batteries with high concentrations of ZnI_2 exhibit greater polarization voltage and lower charge discharge efficiency. On the other hand, the lower ion concentrations than 0.2 M will affect the mass loading of I_2 and capacity of the battery. Therefore, 0.2 M is selected as the optimal concentration for subsequent experiments.

(Page 48 of the revised SI)

1. *The title of the manuscript and the supporting information do not match. Please verify and ensure consistency across both documents.*

Response: We are sorry for this mistake, and thank you very much for the suggestion. We have modified the title in the supporting information.

Changes:

The title of the supporting information: **A tripartite synergistic optimization strategy with cathode host, electrolyte additive and *in-situ* anode protection layer towards superior zinc-iodine batteries.** (Page 1 of the revised SI)

2. *The manuscript extensively discusses the impact of the electrolyte composition, specifically the addition of ZnI_2 and *n*-butanol, on the zinc anode. However, it lacks an explanation of how electrolyte optimization could address critical issues in Zn- I_2 batteries, such as the shuttling of polyiodide ions.*

Response: We appreciate the valuable comment. The use of *n*-butanol additives to address the problem of I_3^- shuttle can be mainly divided into two aspects: (1) inhibiting the dissolution of I_2 , thereby reducing the production of I_3^- ; and (2) utilizing its strong adsorption ability for I_3^- to fix the generated I_3^- in the cathode region. Optical photographs, *ex-situ* UV, and Raman spectroscopy have been conducted to analyze the role of electrolyte optimization.

Firstly, the photos were taken when the ZSI and ZSI-*n* systems were fully charged (Supplementary Fig. 19). The yellow substance (I_3^-) produced in the cathode region of

the ZSI-*n* system is significantly less than that in the ZSI system, indicating that the addition of *n*-butanol reduces the production of I₃⁻. Additionally, the anode region of the ZSI-*n* system is colorless, while the anode region of the ZSI system has a distinct yellow color, indicating that *n*-butanol can effectively fix I₃⁻ ions in the cathode region, limiting its shuttle to the anode.

Ex-situ UV and Raman tests are further conducted on the electrolytes in the cathode and anode regions, respectively. For the ZSI electrolyte without *n*-butanol additive (blue curve in Supplementary Fig. 20), the content of I₃⁻ ions in the electrolyte in the cathode region continues to increase during charging process (Supplementary Figs. 20a-c), indicating the generation I₃⁻ ions. However, as shown in Supplementary Figs. 20d-f, the content of I₃⁻ ions in the anode region electrolyte decreases as the charging process progresses, indicating that I₃⁻ ions are moving back from the anode region to the cathode region. From the Raman spectrum (Supplementary Fig. 5), it can also be seen that in the fully charged state, the peak of I₃⁻ ions at 113 cm⁻¹ is highest in the cathode region and lowest in the anode region. While in the fully discharged state, it is lowest in the cathode region and highest in the anode region (*Adv. Energy Mater.* **2023**, 2300922), which is consistent with the UV results.

After adding the *n*-butanol additive, as shown in the UV spectrum (Supplementary Fig. 20), there is no significant production of I₃⁻ in the cathode region during continuous charging. Besides, in the Raman spectrum (Supplementary Fig. 21), it is found that at different charging and discharging stages and electrode regions, the peaks of I₃⁻ ions (113 cm⁻¹) with *n*-butanol added are lower than that without *n*-butanol, indicating that the *n*-butanol can inhibit the formation of I₃⁻ ions. Additionally, there is no significant change in the concentration of I₃⁻ in the anode region, indicating that *n*-butanol can prevent the shuttle of I₃⁻.

And we have added these data and discussions to the manuscript and supporting information (Note 5, Supplementary Fig. 5, and Supplementary Figs. 19-21).

Changes:

The excellent ability of *n*-butanol to inhibit I₃⁻ shuttle is not only due to its strong adsorption affinity with I₃⁻, but also because it inhibits iodine dissolution, thereby

further reducing the formation of I_3^- . The appeal statement is further confirmed by *ex-situ* UV-vis and Raman spectroscopy, and the corresponding discussions and data are shown in Note 5, Supplementary Fig. 5 and Supplementary Figs. 19-21. (Page 14 of the revised Manuscript)

Supplementary Fig. 19 Photos of (a) ZSI-*n* and (b) ZSI systems in fully charged state (The left electrode is cathode and the right electrode is anode in these two photos). (Page 20 of the revised SI)

Supplementary Fig. 20 The UV spectrum of ZSI and ZSI-*n* electrolyte at cathode side under (a) 0.2 V, (b) 1.3 V, (c) 1.8 V and anode side under (d) 0.2 V, (e) 1.3 V, (f) 1.8 V. (Page 21 of the revised SI)

Supplementary Fig. 5 The Raman spectrum of ZSI electrolyte at (a) cathode side and (b) anode side under different charge states (The illustrations are partial enlarged views).

(Page 6 of the revised SI)

Supplementary Fig. 21 The Raman spectrums of ZSI and ZSI-*n* electrolyte at cathode side under (a) 0.2 V, (b) 1.3 V, (c) 1.8 V and anode side under (d) 0.2 V, (e) 1.3 V, (f) 1.8 V. (Page 22 of the revised SI)

Note 5: The inhibitory effect of *n*-butanol on I_3^- ions shuttle is further confirmed by experiments. Firstly, the photos of ZSI and ZSI-*n* systems are recorded when they are fully charged (Supplementary Fig. 19). The yellow substance (I_3^- ions) produced in the cathode region of the ZSI-*n* system is significantly less than that in the ZSI system, indicating that the addition of *n*-butanol reduces the production of I_3^- ions. Additionally,

the anode region of the ZSI-*n* system is colorless, while the anode region of the ZSI system has a distinct yellow color, indicating that *n*-butanol can effectively fix I_3^- ions in the cathode region, limiting its shuttle to the anode.

Ex-situ UV and Raman tests are further conducted on the electrolytes in the cathode and anode regions, respectively. For the ZSI electrolyte without the *n*-butanol additive (blue curve in Supplementary Fig. 20), the content of I_3^- ions in the electrolyte in the cathode region continues to increase during charging process (Supplementary Figs. 20a-c), indicating the generation I_3^- ions. However, as shown in Supplementary Figs. 20d-f, the content of I_3^- ions in the anode region electrolyte decreases as the charging process progresses, indicating that I_3^- ions are moving back from the anode region to the cathode region. From the Raman spectrum (Supplementary Fig. 5), it can also be seen that in the fully charged state, the peak of I_3^- ions at 113 cm^{-1} is highest in the cathode region and lowest in the anode region. While in the fully discharged state, the peak of I_3^- ions is lowest in the cathode region and highest in the anode region⁹, which is consistent with the UV results.

After adding the *n*-butanol additive, as shown in the UV spectrum (Supplementary Fig. 20), there is no significant production of I_3^- ions in the cathode region during continuous charging. Besides, in the Raman spectrum (Supplementary Fig. 21), it is found that at different charging and discharging stages and electrode regions, the peaks of I_3^- ions (113 cm^{-1}) with *n*-butanol added are lower than that without *n*-butanol, indicating that the *n*-butanol can inhibit the formation of I_3^- ions. Additionally, there is no significant change in the concentration of I_3^- ions in the anode region, indicating that *n*-butanol can prevent the shuttle of I_3^- ions. (Page 51 of the revised SI)

3. *The experimental section notes that the cathode does not contain I_2 , and the only source of iodide (I^-) is from the ZnI_2 additive. Could you specify the active material mass in the battery system? Given that ZnI_2 is used in a small amount, how does this affect the energy density of the battery? An analysis or discussion on this aspect would enhance the understanding of the performance of the battery.*

Response: We thank the reviewer for the valuable comment. First, we would make a correction that the concentration of ZnI_2 is 0.2 M instead of 0.02 M as mentioned above. As shown in Table 1, the mass loading of I_2 and areal energy density of the battery are as high as 1.45 mg cm^{-2} and 0.34 mWh cm^{-2} , respectively, which are higher than most current Zn- I_2 batteries.

Here we use the MXene host without active I_2 loading due to its unique advantages. Reported methods for I_2 loading generally fall into two categories: (1) mixing I_2 with conductive agents and adhesives and coating them onto a substrate, and (2) placing the cathode electrode host in an iodine vapor atmosphere to load I_2 through the host's adsorption capacity. The main disadvantage of the first method is significant I_2 loss during drying and insufficient adsorption by the adhesive, leading to rapid capacity decay during cycling. Although the second method can address weak adsorption capacity, relying solely on the host's adsorption can result in uneven iodine distribution and damage to the channels (*Adv. Mater.* **2021**, *33*, 2006897). Compared with these two methods, the *in-situ* electrodeposition in this study enables iodine to grow uniformly on the surface and within MXene, enhancing its conductivity and ion diffusion ability. More importantly, the amount of I_2 on the cathode prepared by the *in-situ* electrodeposition method can be flexibly adjusted according to the concentration of I^- in the electrolyte. In this way, the mass loading of active iodine on the electrodes can exceed that of the general loading methods.

In addition, the concentration of ZnI_2 is for sure will significantly affect the amount of I_2 and energy density of the battery. As the reviewer pointed out, all the I_2 produced on the cathode material come from ZnI_2 in electrolyte in this study. A small amount of ZnI_2 will lead to a decrease in energy density due to the limited amount of I_2 that can be generated. However, this does not mean that the higher concentration of ZnI_2 , the better performance of battery. We find that too high concentrations of ZnI_2 are more likely to combine with the generated I_2 to form I_3^- , resulting in severe shuttle effects. Furthermore, the too high concentrations of ZnI_2 can reduce the lifespan of zinc anode.

According to your suggestion, we have added corresponding discussions and data in the revised manuscript and supporting information (Note 6, Supplementary Fig. 42, Note 2, and Supplementary Figs. 8-9).

Changes:

It should be noted that although there is no I_2 added in the initial full cells, the I^- in the electrolyte can be converted to I_2 deposited on the MXene host during charging process. The *in-situ* electrodeposition enables I_2 to grow uniformly on the surface within MXene, enhancing its conductivity and ion diffusion ability⁸. More importantly, in fully charge state, the mass loading of I_2 is as high as 1.45 mg cm^{-2} (calculation methods and data in Note 6, Supplementary Fig. 42), which is higher than most reported similar batteries (Table 1). (Page 25 of the revised Manuscript)

It should be noted that the concentration of ZnI_2 (0.2 M) was selected through experimental exploration, and the corresponding discussion and data are presented in Note 2 and Supplementary Figs. 8-9. (Page 10 of the revised Manuscript)

Supplementary Fig. 42 TG curve of the cathode electrode in ZSI-n. (Page 43 of the revised SI)

Supplementary Fig. 8 Capacity and Coulombic efficiency of different contents of ZnI₂ (2 M ZnSO₄+0.2 M ZnI₂, 2 M ZnSO₄+0.5 M ZnI₂ and 2 M ZnSO₄+1.0 M ZnI₂) (Page 9 of the revised SI)

Supplementary Fig. 9 Zn//Zn symmetrical tests of different contents of ZnI₂ (2 M ZnSO₄+0.2 M ZnI₂, 2 M ZnSO₄+0.5 M ZnI₂ and 2 M ZnSO₄+1.0 M ZnI₂) (Page 10 of the revised SI)

Note 6: Unlike most reports, where I₂ is loaded directly on the cathodes prior to battery assembly, the I₂ is *in-situ* electrodeposited on cathode from I⁻ ions during charging process in this study. Here, the mass loading of I₂ on the cathode can be analyzed through TG test³⁹. As shown in the Supplementary Fig. 42, the initial weight loss is caused by the loss of adsorbed water, and the subsequent weight loss until 247 °C is due to the loss of I₂, with an I₂ content of 1.45 mg cm⁻². In addition, the sustained weight loss from 247 °C to 500 °C is the thermal decomposition of PVDF^{40,41}. (Page 52 of the revised SI)

Note 2: Three electrolytes with different contents of ZnI₂ (2 M ZnSO₄+0.2 M ZnI₂, 2 M ZnSO₄+0.5 M ZnI₂ and 2 M ZnSO₄+1.0 M ZnI₂) were prepared for long cycle testing and Zn//Zn symmetric testing. From the Supplementary Fig. 8, it can be seen that electrolytes containing different concentrations of ZnI₂ have little effect on capacity, and even the capacity decreases in electrolytes containing high concentrations of ZnI₂. Besides, the battery with an electrolyte containing 0.2 M ZnI₂ exhibits a high Coulombic efficiency of 93%, which is much higher than the other two electrolytes (0.5 M ZnI₂: 12%; 1.0 M ZnI₂: 2%). This is because high concentrations of I⁻ are more likely to combine with I₂ to form I₃⁻, leading to overcharging and the production of more I₃⁻, resulting in severe shuttle effects. In addition, the effect of different concentrations of ZnI₂ on the zinc anode is further investigated. As shown in the Supplementary Fig. 9, symmetric batteries with high concentrations of ZnI₂ exhibit greater polarization voltage and lower charge discharge efficiency. On the other hand, the lower ion concentrations than 0.2 M will affect the mass loading of I₂ and capacity of the battery. Therefore, 0.2 M is selected as the optimal concentration for subsequent experiments.

(Page 48 of the revised SI)

Table 1. Summary of the optimization strategy, load mass, current density, area capacity density, area energy density and cycle number of Zn-I₂ batteries reported recently. (Page 45 of the revised SI)

Reference	Optimization strategy	Load mass (mg cm ⁻²)	Current density (A g ⁻¹)	Area capacity density (mAh cm ⁻²)	Area energy density (mWh cm ⁻²)	Cycle number
[1]	Cathode	0.06	1	0.014	0.016	1500
[2]	Cathode Anode	1.0	2.5	0.118	0.13	3000
[3]	Cathode	1.0	2.11	0.175	0.21	10000
[4]	Anode	0.63	2	0.126	0.15	10000
[5]	Electrolyte additive In-situ SEI	1.38	1.45	0.259	0.30	5000
[6]	Cathode	1.0	5	0.095	0.11	20000
[7]	Anode	1.3	2	0.135	0.15	20000

[8]	Anode	1.0-2.0	3.2	0.09-0.17	0.10-0.20	6000
[9]	Cathode	1.0	1	0.2	0.25	5700
[10]	Cathode	1.0	4.22	0.14	0.16	10000
[11]	Cathode	0.8-1.5	2	0.16-0.3	0.19-0.35	10000
[12]	Cathode	1.25	4	0.25	0.28	1500
[13]	Gel electrolyte	1.5	1.58	0.24	0.26	5000
[14]	Cathode	2	5	0.24	0.29	5000
[15]	Cathode	2	8.44	0.16	0.19	6000
[16]	Cathode	1.0-1.2	2.11	0.15-0.18	0.16-0.20	20000
[17]	Cathode	1.17	4.22	0.154	0.20	5000
[18]	Cathode	1.0-1.2	1.2	0.21-0.25	0.27-0.32	1000
[19]	Cathode	1.0-1.2	5	0.16-0.19	0.19-0.22	30000
[20]	Gel electrolyte	0.8-1.0	2.11	0.12-0.15	0.14-0.17	22000
[21]	Ionic liquid electrolyte	0.8-1.0	4	0.12-0.15	0.14-0.17	18000
[22]	Cathode	1.0-1.2	5	0.17-0.2	0.20-0.24	17000
[23]	Cathode	1.0-2.0	2	0.1-0.2	0.12-0.23	1400
[24]	Cathode	0.8-1.2	3	0.1-0.15	0.12-0.18	5900
This work	Cathode Electrolyte additive In-situ SEI	1.45	5	0.30	0.34	30000

The corresponding references are listed in the supporting information

4. The manuscript highlights the issue of I_3^- ion shuttling in Zn- I_2 batteries, yet it provides insufficient details on how I_3^- shuttles between electrodes. A more thorough explanation of this shuttle mechanism is necessary to contextualize the discussion properly.

Response: Thank you very much for your comment. The mechanism of I_3^- shuttle is to move from anode to cathode during charging and from cathode to anode during discharging (*Energy Environ. Sci.* **2023**, *16*, 4630). Here *ex-situ* UV and Raman spectroscopy were conducted to further analyze the I_3^- shuttles.

Thermodynamically, the I_2 is able to combine with I^- ions to form I_3^- ions (*Energy Environ. Sci.* **2017**, *10*, 735-741; *Nano Lett.* **2021**, *21*, 4129-4135). Thus, the continuous

production of I_3^- ions in the electrolyte near the cathode creates a concentration gradient, promoting the diffusion of I_3^- towards the anode. As shown in Supplementary Fig. 4, when charged to 1.8V for the first time (the red line), a large amount of I_3^- ions are generated in the cathode region electrolyte, with a small amount also appearing in the anode region. This indicates that I_3^- ions move from the cathode to the anode during the charging process due to the concentration gradient. During the discharging process, under the influence of an external potential, the anions in the electrolyte move to the anode. As shown by the red and gray curves in Supplementary Fig. 4, the concentration of I_3^- ions in the anode region gradually increases during the discharging process. In the subsequent charging process, I_3^- ions move back to the cathode region due to the electric field. From the Raman spectrum (Supplementary Fig. 5), it can also be seen that in the fully charged state, the peak of I_3^- at 113 cm^{-1} is highest in the cathode region and lowest in the anode region. While in the fully discharged state, the peak of I_3^- is lowest in the cathode region and highest in the anode region (*Adv. Energy Mater.* **2023**, 2300922), which is consistent with the UV results. Therefore, the mechanism of the I_3^- shuttle involves its movement from the anode to the cathode during charging and from the cathode to the anode during discharging, similar with the previous reports (*Energy Environ. Sci.* **2023**, 16, 4630).

According to your suggestion, we have included the corresponding results in the revised manuscript and supporting information (Note 4, Supplementary Figs. 4-5).

Changes:

In order to more intuitively reflect this role of *n*-butanol, the mechanism of I_3^- shuttle is studied and discussed in Note 4 and Supplementary Figs. 4-5. (Page 14 of the revised Manuscript)

Supplementary Fig. 4 The UV spectrum of ZSI electrolyte at (a) cathode side and (b) anode side under different charge states. (Page 5 of the revised SI)

Supplementary Fig. 5 The Raman spectrum of ZSI electrolyte at (a) cathode side and (b) anode side under different charge states (The illustrations are partial enlarged views). (Page 6 of the revised SI)

Note 4: Thermodynamically, the I_2 is able to combine with I^- ions to form I_3^- ions^{37,38}. Thus, the continuous production of I_3^- ions in the electrolyte near the cathode creates a concentration gradient, promoting the diffusion of I_3^- towards the anode. As shown in Supplementary Fig. 4, when charged to 1.8V for the first time (the red line), a large amount of I_3^- ions are generated in the cathode region electrolyte, with a small amount also appearing in the anode region. This indicates that I_3^- ions move from the cathode to the anode during the charging process due to the concentration gradient. During the discharging process, under the influence of an external potential, the anions in the electrolyte move to the anode. As shown by the red and gray curves in Supplementary Fig. 4, the concentration of I_3^- ions in the anode region gradually increases during the

discharging process. In the subsequent charging process, I_3^- ions move back to the cathode region due to the electric field. From the Raman spectrum (Supplementary Fig. 5), it can also be seen that in the fully charged state, the peak of I_3^- at 113 cm^{-1} is highest in the cathode region and lowest in the anode region. While in the fully discharged state, the peak of I_3^- is lowest in the cathode region and highest in the anode region⁹, which is consistent with the UV results. Therefore, the mechanism of the I_3^- ions shuttle involves its movement from the anode to the cathode during charging and from the cathode to the anode during discharging, similar with the previous reports⁷. (Page 50 of the revised SI)

5. The manuscript asserts that the TMX cathode host, characterized by its layered structure and surface-rich groups, facilitates the I/I_2 conversion reaction, which purportedly reduces I_3^- formation. However, this claim lacks empirical backing. It is advisable to include *in situ* or *ex situ* Raman spectroscopy, XRD, or XPS data to provide evidence of the iodine conversion mechanism.

Response: We thank for your valuable comment. The layered structured Ti_3C_2Tx has abundant exposed and unsaturated Ti atoms, which has strong absorbability and can act as the active sites for redox reactions (*Energy Storage Mater.* **2020**, 25, 885-892). In addition, the oxygen-terminated MXene surface is endowed with high catalytic activity toward conversion reaction (*Adv. Energy Mater.* **2019**, 9, 1900219). Furthermore, we have studied the mechanism of iodine conversion reaction through *ex-situ* UV, Raman, XPS and XRD. In summary, the following conversion reactions occur on the cathode electrode during the charging and discharging process. During charging, I^- ions lose electrons and convert to I_2 , which is be able to combine with I^- ions to form I_3^- ions. During discharge, I_2 or I_3^- ions get electrons and convert into I^- . According to your suggestion, we have added the experimental process and discussions in the revised manuscript and supporting information (Note.1, Supplementary Figs. 4-7).

Discharging: $I_2/I_3^- + 2e^- \rightarrow 2I^-/3I^-$

Charging: $2I^-/3I^- - 2e^- \rightarrow I_2/I_3^-$

Changes:

In addition, the mechanism of iodine conversion reaction on TMX is described in Note 1 and Supplementary Figs. 4-7. (Page 10 of the revised Manuscript)

Supplementary Fig. 4 The UV spectrum of ZSI electrolyte at (a) cathode side and (b) anode side under different charge states. (Page 5 of the revised SI)

Supplementary Fig. 5 The Raman spectrum of ZSI electrolyte at (a) cathode side and (b) anode side under different charge states (The illustrations are partial enlarged views). (Page 6 of the revised SI)

Supplementary Fig. 6 The XPS of cathode electrode under different charge states.

(Page 7 of the revised SI)

Supplementary Fig. 7 The XRD of cathode electrode under different charge states.

(Page 8 of the revised SI)

Note 1: The mechanism of iodine conversion reaction on TMX is further analyzed in this study. First, *ex-situ* UV test on the electrolyte in the cathode region is conducted, and the results are shown in the Supplementary Fig. 4a. During the charging process, the concentration of I_3^- in the electrolyte near the cathode region continues to increase, indicating that I_2 continues to be generated on the cathode electrode and combines with I^- to form I_3^- ³⁶. The peak located at 113 cm^{-1} in Raman testing also continues to increase during the charging process (Supplementary Fig. 5a), indicating an increase of I_3^- , which is consistent with the UV results. Moreover, XPS test is conducted on the cathode (Supplementary Fig. 6), and it is found that the peak intensities of I_3^- located at 632.36

and 620.54 eV continue to increase during the charging process, indicating the continuous generation of I_3^- , also consistent with the UV test results. The peaks of I_2 (631.87 and 620.15 eV) increase during the charging process, while the peaks of I^- (631.13 and 619.64 eV) decrease, indicating that I^- ions are continuously converted into I_2 and loaded on the cathode during the charging process^{4,9}. XRD test is further conducted on the cathode under different charge states (Supplementary Fig. 7). Compared with the initial cathode, the fully discharged cathode shows significant peak enhancement at 25.50° and 42.50°, which corresponding to ZnI_2 (PDF#10-0072). Besides, there is no obvious I_2 peak, indicating that most of the I_2 is reduced to I^- ions in the fully discharged state¹⁴. When fully charged to 1.8V, the peak at 25.50° is almost indistinguishable, while a clear peak corresponding to I_2 appears at 24.47° (PDF#43-0304), indicating that most I^- ions are oxidized to I_2 during the charging process. In summary, the following conversion reactions occur on the cathode during the charging and discharging process. During charging, the electrons of I^- are converted into I_2 , which combines with I^- to form I_3^- . During discharging, I_2 or I_3^- lose electrons and convert into I^- .

6. In Figure 5f, the authors present a comparison of cycle number, current density, and capacity among various reported Zn- I_2 batteries. However, specific references for the comparative data are missing. Please provide these references and the comparative data for further reading. It is noted that the paper uses additives as active materials, which qualitatively differ from the materials typically used in iodine cathodes. Since iodine cathodes generally contain a higher mass of active substances, comparing these different systems may not provide a meaningful analysis. Please reconsider the basis for this comparison or provide a detailed rationale for including such disparate systems in the comparative analysis to ensure the validity and relevance of the discussion.

Response: We thanks for the valuable comments and we will answer these questions one by one.

Firstly, the reviewer noted that our materials “the qualitatively in different from the materials typically used in iodine cathodes.” In fact, the I₂ loaded on the cathode directly is essentially similar with the *in-situ* electrodeposited I₂ in this study. If I₂ is loaded on the cathode directly, the assembled Zn-I₂ battery first need to be discharged, reducing the I₂ on the cathode to I⁻ ions, which will diffuse into electrolyte. In this work, the fabricated Zn-I₂ battery first need to be charged, oxidizing the I⁻ ions to I₂ on the cathode. In the subsequent charging and discharging process, the redox reaction that occurs in both types of Zn-I₂ batteries is the same.

Secondly, we use MXene as the cathode electrode host without an active substance and employ an *in-situ* electrodeposition method to load iodine. This enables iodine to grow uniformly on the surface and within MXene, enhancing its conductivity and ion diffusion ability. Reported methods for iodine loading generally fall into two categories: (1) mixing iodine with conductive agents and adhesives and coating it onto a substrate, and (2) placing the cathode electrode host in an iodine vapor atmosphere to load iodine through the host's adsorption capacity. The main disadvantage of the first method is significant iodine loss during drying and insufficient adsorption by the adhesive, leading to rapid capacity decay during continuous cycling. In our work, iodine generated by *in-situ* electrodeposition does not suffer from temperature-related loss and relies on the MXene cathode host and n-butanol to effectively fix iodine. While the second method can address weak adsorption capacity, relying solely on the host's adsorption can result in uneven iodine distribution and damage to the channels (*Adv. Mater.* **2021**, *33*, 2006897).

Thirdly, the reviewer mentioned that “iodine cathodes generally contain a higher mass of active substances, comparing these different systems may not provide a meaningful analysis.” Thus, we obtained the mass loading of I₂ on cathode after *in-situ* electrodeposition by TG measurement (as shown in Note 6 and Supplementary Fig. 42), and compared with other reported Zn-I₂ batteries in recent years. As shown in Table 1, the mass loading of I₂ the battery is as high as 1.45 mg cm⁻², which is higher than most current Zn-I₂ batteries.

Finally, regarding the reviewer's mention of “missing specific references for the comparative data,” we have supplemented more specific data, including the mass loading of I₂, area capacity, area energy density and cycle number. Furthermore, we conduct a comparison of these important properties with other similar batteries (Table 1), and our battery demonstrates advantages in multiple aspects.

Changes:

Table 1. Summary of the optimization strategy, load mass, current density, area capacity density, area energy density and cycle number of Zn-I₂ batteries reported recently. (Page 45 of the revised SI)

Reference	Optimization strategy	Load mass (mg cm ⁻²)	Current density (A g ⁻¹)	Area capacity density (mAh cm ⁻²)	Area energy density (mWh cm ⁻²)	Cycle number
[1]	Cathode	0.06	1	0.014	0.016	1500
[2]	Cathode Anode	1.0	2.5	0.118	0.13	3000
[3]	Cathode	1.0	2.11	0.175	0.21	10000
[4]	Anode	0.63	2	0.126	0.15	10000
[5]	Electrolyte additive In-situ SEI	1.38	1.45	0.259	0.30	5000
[6]	Cathode	1.0	5	0.095	0.11	20000
[7]	Anode	1.3	2	0.135	0.15	20000
[8]	Anode	1.0-2.0	3.2	0.09-0.17	0.10-0.20	6000
[9]	Cathode	1.0	1	0.2	0.25	5700
[10]	Cathode	1.0	4.22	0.14	0.16	10000
[11]	Cathode	0.8-1.5	2	0.16-0.3	0.19-0.35	10000
[12]	Cathode	1.25	4	0.25	0.28	1500
[13]	Gel electrolyte	1.5	1.58	0.24	0.26	5000
[14]	Cathode	2	5	0.24	0.29	5000
[15]	Cathode	2	8.44	0.16	0.19	6000
[16]	Cathode	1.0-1.2	2.11	0.15-0.18	0.16-0.20	20000
[17]	Cathode	1.17	4.22	0.154	0.20	5000
[18]	Cathode	1.0-1.2	1.2	0.21-0.25	0.27-0.32	1000
[19]	Cathode	1.0-1.2	5	0.16-0.19	0.19-0.22	30000
[20]	Gel electrolyte	0.8-1.0	2.11	0.12-0.15	0.14-0.17	22000

[21]	Ionic liquid electrolyte	0.8-1.0	4	0.12-0.15	0.14-0.17	18000
[22]	Cathode	1.0-1.2	5	0.17-0.2	0.20-0.24	17000
[23]	Cathode	1.0-2.0	2	0.1-0.2	0.12-0.23	1400
[24]	Cathode	0.8-1.2	3	0.1-0.15	0.12-0.18	5900
This work	Cathode Electrolyte additive In-situ SEI	1.45	5	0.30	0.34	30000

The corresponding references are listed in the supporting information.

Reviewer #2 (Remarks to the Author):

The manuscript presents a tripartite synergistic optimization strategy involving a cathode host, electrolyte additive, and in-situ formed zinc anode protective layer. This approach addresses challenges related to iodine dissolution and zinc dendrite growth in zinc-iodine batteries, resulting in exceptional electrochemical performance characterized by high capacity and extended cycle life. The authors also elucidate the charge storage mechanisms using those techniques. Overall, this paper holds promise for publication in Nature Communications following some revisions. Below are the detailed comments for the authors:

Response: We highly appreciate the reviewer for the positive comments. We have read all the comments very carefully and revised the manuscript accordingly.

1. The area capacity in ZSI electrolyte is reported as 0.22 mAh cm⁻², while introducing the n-butanol additive increases it to 0.30 mAh cm⁻². Please explain the rationale behind this enhancement in area capacity upon adding the electrolyte additive.

Response: We thank the reviewer for the valuable comment. The capacity improvement brought about by the introduction of n-butanol electrolyte additives can be mainly attributed to the inhibition of the shuttle effect of I₃⁻.

In our work, we use an MXene cathode host without active substance, meaning that all capacity contributions come from the iodine source in the electrolyte. During the charging process, I⁻ ions on the cathode electrode lose their electrons and convert into I₂, which can contribute capacity in subsequent discharging process. At the same time, the generated I₂ is also able to combine with the I⁻ ions to form I₃⁻ (Equation 1, *Chem. Rev.* **2003**, *103*, 5, 1649-1684). Due to the concentration gradient, the large amount of I₃⁻ on the cathode electrode spontaneously passes through the separator and reaches the zinc anode electrode. Since the standard redox potential of I₃⁻/I⁻ (0.5355 vs. SHE) is higher than that of Zn²⁺/Zn (-0.763 vs. SHE), the Zn metal can be oxidized by I₃⁻, leading to the corrosion of the Zn anode and the loss of the iodine source. This is

the detrimental shuttle effect of I_3^- (*Adv. Energy Mater.* **2023**, *13*, 2302187). In brief, less I_3^- shuttle results in higher capacity and cycling stability.

When the ZSI and ZSI-*n* systems are fully charged (Supplementary Fig. 19), it can be seen that the anode region of the ZSI-*n* system does not show significant yellow color (I_3^-). While the anode region of the ZSI system shows a distinct yellow color, indicating no shuttle phenomenon of I_3^- in the ZSI-*n* system. Furthermore, the UV tests of the electrolytes in the cathode and anode regions (Supplementary Fig. 20) reveal that the concentration of I_3^- in both regions of the ZSI-*n* system is much lower than that in the ZSI system, indicating that the *n*-butanol can effectively fix I_3^- and restricts its shuttle.

We have also added the corresponding discussions and data to the revised manuscript and supporting information.

Changes:

The capacity improvement brought by the introduction of *n*-butanol electrolyte additives can be mainly attributed to the inhibition of the shuttle effect of I_3^- . (Page 26 of the revised Manuscript)

Supplementary Fig. 19 Photos of (a) ZSI-*n* and (b) ZSI systems in fully charged state (The left electrode is cathode and the right electrode is anode in these two photos). (Page 20 of the revised SI)

Supplementary Fig. 20 The UV spectrum of ZSI and ZSI-*n* electrolyte at cathode side under (a) 0.2 V, (b) 1.3 V, (c) 1.8 V and anode side under (d) 0.2 V, (e) 1.3 V, (f) 1.8 V. (Page 21 of the revised SI)

2. The authors determine the thickness of TMX to be 1.9 nm using AFM. How does the thickness impact electrochemical behaviors, and is a thinner TMX preferable for better electrochemical performance?

Response: We thank the reviewer for the valuable comment. $\text{Ti}_3\text{C}_2\text{T}_x$ MXene is a typical 2D nanomaterial with single- or few-layer inorganic 2D nanosheets. The layered structured $\text{Ti}_3\text{C}_2\text{T}_x$ has abundant exposed and unsaturated Ti atoms, which has strong absorbability and can act as the active sites for redox reactions (*Energy Storage Mater.* **2020**, *25*, 885-892). Electrodes fabricated by MXene possess high surface area and edge sites (*Chem. Rev.* **2017**, *117*, 6225-6331), which are beneficial for increasing electrochemical active sites, shortening the diffusion distance of electrolyte ions (*Adv. Energy Mater.* **2017**, *7*, 1602684), leading to the enhanced specific capacity and electrochemical stability (*ACS Nano* **2020**, *14*, 1176-1184; *Chem. Commun.* **2010**, *46*, 1106-1108). For a given loading of TMX, thinner nanosheets mean more layers, which translates to increased interlayer spacing and a larger specific surface area. Therefore, theoretically, thinner TMX nanosheets are more conducive to improving

electrochemical performance.

3. *More experimental details are necessary for result reproducibility. For instance, the diameter and thickness of the zinc electrode, electrolyte volume, and separator diameter.*

Response: We are grateful for the comment. The thickness and area of the zinc electrode are 0.03 mm and 1.13 cm² respectively. The area of the glass-fiber separator is 2.01 cm², and the amount of electrolyte droplets added to each coin cell is 80 μL. In addition, we have also added these details to the manuscript.

Changes:

The thickness and area of the zinc electrode are 0.03 mm and 1.13 cm² respectively. The area of the glass-fiber separator is 2.01 cm², and the amount of electrolyte droplets added to each coin cell is 80 μL. (Page 30 of the revised Manuscript)

4. *The captions corresponding to the three colors in figures 4d and 5e appear to be inaccurate. While an increase in overpotential is expected in symmetric battery testing due to residual SEI accumulation and electrolyte depletion, explain the observed trend of decreasing and then increasing polarization voltage corresponding to the blue and pink colors.*

Response: Thanks very much for this scientific question. We have revised the captions of fig. 4d and 5e and checked all the captions in the manuscript.

The trend of first decreasing is caused by the increased surface area upon repetitive deposition and dissolution. The larger surface area can provide more reactive sites, making the reaction easier to occur. These active sites can adsorb reactants and adjust their electronic structure, thereby reducing the overpotential of the reaction. Similar phenomena have also been observed in other reports as shown in Fig. R2.

Fig. R2. Zn//Zn Symmetric cyclic figures of reported work (a. *Nat. Commun.* **2023**, *14*, 2720; b. *Energy Storage Mater.* **2023**, *61*, 102882; c. *Energy Environ. Sci.* **2023**, *16*, 4572-4583; d. *Adv. Energy Mater.* **2022**, *12*, 2103231).

Changes:

Fig. 4 | Stability and mechanism analysis of zinc anodes in different electrolytes. a Voltage profiles of galvanostatic Zn deposition on Cu at 1 mA cm^{-2} . b CA tests in

different electrolytes. **c** CE profiles of Zn//Cu cells in three electrolytes. **d** Symmetrical cells operating at **f** 1 mA cm^{-2} . **e** Surface energy value for main planes of metallic Zn in three electrolytes. **f** Optical microscopy image of Zn/electrolyte interface during Zn deposition in ZS, ZSI and ZSI-*n*. **g** High-resolution TEM images of the Zn anode surface after cycling in three electrolytes. **h** LCSM images of Zn anodes after cycling in ZSI-*n*. **i** Values of surface adsorption energy of H_2O , SO_4^{2-} , I^- and *n*-butanol on Zn (002) plane. (Page 24 of the revised Manuscript)

Fig. 5 | Electrochemical performance of full cell with MXene as cathode host, ZS/ZSI/ZSI-*n* as electrolyte and zinc as anode. a CV curves of ZS, ZSI and ZSI-*n* at 0.4 mV s^{-1} . Galvanostatic charge-discharge (GCD) curves of **b** ZS, **c** ZSI, **d** ZSI-*n*. **e** Rate performance comparison of ZS, ZSI and ZSI-*n*. **f** Comparison of cycle number, current density and capacity between our battery and other reported Zn-I₂ batteries. **g** cycling capability of full cell with ZS, ZSI and ZSI-*n* as electrolyte at 5 A g^{-1} . (Page 27 of the revised Manuscript)

5. When discussing the adsorption of TMX to I_3^- using UV spectroscopy, the authors should cite relevant literature, particularly regarding the peaks observed in the UV spectra.

Response: We thank the reviewer for the valuable comment. We have added relevant references at the corresponding discussion section.

Changes:

After a week of soaking, UV spectrum results of these two solutions (Supplementary Fig. 3) indicate significantly weaker I_3^- bands for the solution treated with TMX compared to that treated with CC, showcasing the excellent adsorption effect of TMX on I_3^- ²⁰. Further UV testing on fully charged solutions at 1.8 V reveals a more pronounced concentration difference between the electrolytes with TMX and CC as cathode electrode (Fig. 2c), which confirms the adsorption ability of TMX for I_3^- ²¹.

(Page 9 of the revised Manuscript)

20. Yang, J. et al. Janus Binder Chemistry for Synchronous Enhancement of Iodine Species Adsorption and Redox Kinetics toward Sustainable Aqueous Zn-I₂ Batteries. *J. Am. Chem. Soc.* **146**, 6628-6637 (2024).

21. Sonigara K. et al. Self-Assembled Solid-State Gel Catholyte Combating Iodide Diffusion and Self-Discharge for a Stable Flexible Aqueous Zn-I₂ Battery. *Adv. Energy Mater.* **10**, 2001997 (2020). (Pages 36-37 of the revised Manuscript)

6. Please provide corresponding equivalent circuits to fit the Nyquist diagram in the supplementary information section.

Response: We are grateful for the suggestion. In response, we have included an equivalent circuit diagram to fit the Nyquist diagram in Supplementary Fig. 11. In addition, the equivalent circuit diagram has also been used for fitting the EIS curves presented in the Supplementary Figs. 27, 29, and 43.

Changes:

Supplementary Fig. 11 EIS curves of CC and TMX. (The illustration shows the corresponding equivalent circuit) (Page 12 of the revised SI)

Supplementary Fig. 27 CA curve and the corresponding EIS plots of the symmetric cell with (a) ZS, (b) ZSI and (c) ZSI-*n* electrolytes. (The corresponding equivalent circuit is consistent with Supplementary Fig. 11) (Page 28 of the revised SI)

Supplementary Fig. 29 The EIS impedance of Zn//Zn battery in (a) ZS, (b) ZSI and (c) ZSI-*n* electrolytes at different temperature. (The corresponding equivalent circuit is consistent with Supplementary Fig. 11) (Page 30 of the revised SI)

Supplementary Fig. 43 EIS curves of ZS, ZSI and ZSI-*n*. (The corresponding equivalent circuit is consistent with Supplementary Fig. 11) (Page 44 of the revised SI)

7. Please explain the potential advantages of suspended *n*-butanol over fully dissolved one. Elucidate the underlying principles supporting this distinction.

Response: We thank the reviewer for the valuable comment. In our work, *n*-butanol forms a suspended electrolyte due to its low solubility in aqueous solution, offering several advantages over fully dissolved electrolytes:

- (1) The suspended *n*-butanol in the cathode electrode region effectively fixes I_3^- ions, which helps to mitigate the shuttle effect of I_3^- ions, thereby preventing the loss of the iodine source and corrosion of the anode electrode.
- (2) Partially dissolved *n*-butanol alters the solvation environment of Zn^{2+} , promoting the desolvation of solvated Zn^{2+} and enhancing the transport capacity of Zn^{2+} , which aids in zinc deposition.
- (3) *N*-butanol in the electrolyte encourages zinc to preferentially grow along the (002) crystal plane, resulting in stable and uniform zinc deposition.
- (4) The suspended *n*-butanol additive generates a hydrophobic SEI with inorganic substances on the zinc anode, which inhibits zinc dendrite growth and improves the reversibility and stability of the zinc anode.

In addition to the advantages demonstrated in our work, previous studies have highlighted other benefits of suspended electrolytes in lithium metal batteries (*Nat. Mater.* **2022**, *21*, 445-454): (1) modify the solvation environment through the interfacial interactions between the Li_2O (the suspended electrolyte in his work) surface and its surrounding solvation shells of the liquid electrolyte; (2) Li_2O creates a weakly

solvating environment by decreasing Li⁺-solvent and increasing Li⁺-anion coordination;
(3) Li₂O attracts fluorinated species and dissociated Li⁺.

8. *Several errors should be corrected in the text, such as “Zinc-iodine” in the abstract section, “seperator” in Figure 1. Ensure references include complete information, including year, volume, and page numbers.*

Response: Thanks for your suggestion. We have examined the manuscript carefully and have corrected the errors.

Changes:

- ◆ The energy industry has taken notice of **zinc-iodine (Zn-I₂) batteries** for their high safety, low cost and attractive energy density. **(Page 2 of the revised Manuscript)**
- ◆ While the current reports have made some progress in improving performance of **Zn-I₂ batteries**, the majority of current research primarily concentrates on enhancing either the cathode electrode, anode electrode, or electrolyte, often overlooking the importance of addressing the other part. **(Page 5 of the revised Manuscript)**
- ◆ “the **Zn//Zn symmetric test serves** as another way to assess the reversibility of the zinc anode.” **(Page 19 of the revised Manuscript)**

Fig. 1 | Schematic illustration of the tripartite synergistic optimization strategy with cathode host, electrolyte additive and *in-situ* anode protection layer. (Page 7 of the revised Manuscript)

◆ **Reference:**

1. Xia, M. T. et al. Hydrogen-bond regulation in organic/aqueous hybrid electrolyte for safe and high-voltage K-ion batteries. *Energy Environ. Sci.* **17**, 1255-1265 (2024).
25. Wu, J. et al. Immobilizing Polyiodides with Expanded Zn^{2+} Channels for High-Rate Practical Zinc-Iodine Battery. *Adv. Energy Mater.* **14**, 2302738 (2023).
26. Wei, F. et al. 2D Mesoporous Naphthalene-Based Conductive Heteroarchitectures toward Long-Life, High-Capacity Zinc-Iodine Batteries. *Adv. Funct. Mater.* **34**, 2310693 (2023).
42. He, J. et al. Synergistic effects of Lewis acid–base and Coulombic interactions for high-performance Zn-I₂ batteries. *Energy Environ. Sci.* **17**, 323-331 (2024).
46. Zhang, K. et al. Precipitated Iodine Cathode Enabled by Trifluoromethanesulfonate

Oxidation for Cathode/Electrolyte Mutualistic Aqueous Zn-I Batteries. *Adv. Mater.* **36**, 2309838 (2023).

Reviewer #3 (Remarks to the Author):

This study proposes a new strategy that comprehensively improves the performance of zinc-iodine batteries from multiple aspects, involving a MXene cathode host, a n-butanol electrolyte additive and the in-situ solid electrolyte interface. The prepared battery exhibits enhanced capacity as well as long lifespan, and the corresponding mechanism has also been studied and calculated. The idea is interesting and on interest for the readers of the journal. Accordingly, this work could be accepted for publication provided the following issues are solved:

Response: Thank you very much for your overall positive assessment of this work. Hopefully, our response below may address your concerns.

1. *The capacity of the battery reaches 0.3 mAh cm⁻² at a high current density of 5 A g⁻¹, which looks good. But what I'm curious about is, will the capacity be larger at lower current densities? For example 1 A g⁻¹.*

Response: According to your suggestion, we have tested the full battery at a lower current density of 1 A g⁻¹. Over the course of 2000 cycles, the battery's capacity decreased from 0.38 mAh cm⁻² to 0.33 mAh cm⁻², resulting in a capacity retention rate of 86.8%. In contrast, at a current density of 5 A g⁻¹, the capacity decreased from 0.32 mAh cm⁻² to 0.30 mAh cm⁻², with a capacity retention rate of 93.7%. It can be seen that the capacity increases very little at lower current densities, which is due to the fast I₂/I⁻ conversion reaction kinetics in this study.

Fig. R3. Capacity and Coulombic efficiency of ZSI-n at 1 and 5 A g⁻¹.

2. More specific measurements should be used to study the protection of the anode electrode by the *in-situ* generated SEI film, such as its corrosion resistance.

Response: We appreciate the valuable comment. We have studied the corrosion resistance of the SEI film formed on the anode surface. Based on the fitting results of the polarization curves shown in Supplementary Fig. 25, the *in-situ* generated SEI film significantly impacts the corrosion behavior of zinc anodes. Specifically, ZSI-*n* exhibits a higher corrosion potential (-0.8413 V vs. Ag/AgCl) and a lower corrosion current density (1.9445 mA cm⁻²) than those of ZSI (-0.9643 V with 4.6451 mA cm⁻²) and ZS (-0.9658 V with 3.4276 mA cm⁻²), indicating an increased corrosion resistance as well as a good resistance to Zn corrosion of ZSI-*n*.

Additionally, we have included the discussion and corresponding data in the revised manuscript and supporting information.

Changes:

Additionally, anti-corrosion capability is essential for the protection and long-term utilization of zinc anode, and the corrosion resistance is obtained from the Tafel curve. According to the fitting results of the polarization curves shown in Supplementary Fig. 25, the battery with ZSI-*n* has a remarkable effect on the corrosion behavior of zinc anode. Specifically, ZSI-*n* exhibits a higher corrosion potential (-0.8413 V vs. Ag/AgCl) and a lower corrosion current density (1.9445 mA cm⁻²) than those of ZSI (-0.9643 V with 4.6451 mA cm⁻²) and ZS (-0.9658 V with 3.4276 mA cm⁻²), indicating an increased corrosion resistance as well as a good resistance to Zn corrosion of ZSI-*n*.

(Page 18 of the revised Manuscript)

3. The author claims that the MXene host and *n*-butanol additive inhibit the shuttle of I_3^- synergistically, so which one contributes more?

Response: We thanks for the comment. In order to investigate which contributes more, the MXene cathode electrode host or the *n*-butanol additive, batteries without *n*-butanol additive (named TMX+ZSI) or the MXene cathode electrode host (named CC+ZSI-*n*) were assembled and tested. From the Fig. R4, it can be seen that at the current density of 5 A g^{-1} , TMX+ZSI system has a capacity of 0.32 mA cm^{-2} and a Coulombic efficiency of 95%, while CC+ZSI-*n* system has a capacity of only 0.25 mA cm^{-2} and a low Coulombic efficiency (70%). The above results indicate that TMX contributes more to limiting the shuttle of I_3^- in the entire battery.

Fig. R4. Capacity and Coulombic efficiency of TMX+ZSI and CC+ZSI-*n* at 5 A g^{-1} .

4. There have been several reports on alcohol electrolyte additives before (for example, *Energy Storage Mater.* 48 (2022) 192-204; *Angew. Chem. Int. Ed.* 2021, 60, 7366-7375; *Adv. Funct. Mater.* 2023, 2214538). What is the advantage of *n*-butanol additives compared to previous reports?

Response: In order to protect the zinc anode in ZIBs, several alcohol additives (ethylene glycol, poly(ethyleneglycol), ethanol, methanol, sorbitol, propylene glycol, tripropylene glycol, Polyethylene Glycol, inositol) were added to the electrolyte. These alcohol additives are easily dissolved in aqueous electrolyte, and can regulate the

dissolved structure of Zn^{2+} ions, thus inhibiting the corrosion of the zinc anode. However, for zinc-iodine batteries, in addition to zinc anode problems, the shuttle of I_3^- ions is also an important issue, which cannot be solved with these reported alcohol additives.

In our study, we prepared a special *n*-butanol additive and investigated its working mechanism. Due to the low solubility of *n*-butanol in aqueous solutions, the *n*-butanol additive forms a suspended electrolyte. The suspended electrolyte effectively fixes I_3^- near the cathode electrode due to *n*-butanol's strong adsorption ability, thereby addressing the I_3^- shuttle issue. In contrast, most alcohol additives dissolve well in aqueous solutions and lack the adsorption capacity to inhibit I_3^- shuttle effectively. Additionally, the suspended electrolyte provides robust protection for the anode electrode. As shown in Table 2, our work demonstrates advantages over other studies using alcohol electrolytes, both in extending zinc anode lifespan and in-depth exploration of the additive's working principles. Therefore, the *n*-butanol suspension electrolyte better meets the needs of zinc-iodine batteries compared to other alcohol additives.

Moreover, we have included these discussions and relevant data in the manuscript and supporting information.

Changes:

Furthermore, numerous alcohols serve as electrolyte additives to alter the solvation structure and enhance the lifespan of zinc anode electrodes (as shown in Table 2)^{14,30}, but they are powerless to solve the problem of I_3^- shuttle. Therefore, we are committed to exploring electrolyte additives that can simultaneously solve these problems. (Page 13 of the revised Manuscript)

Table 2. Summary of Zn//Zn symmetrical testing performance and function mechanism of alcohol electrolyte additives. (Page 46-47 of the revised SI)

Reference	Electrolyte	Current density (mA cm ⁻²)	Areal capacity (mAh cm ⁻²)	Lifespan (h)	Function mechanism
[25]	ZnSO ₄ + EG	2	1	150	Electrolyte optimization

[26]	Zn(TFSI) ₂ +LiTFSI +PEG	0.6	0.1	450	Electrolyte optimization; Surface film adjustment
[27]	NaTfO+Zn(TfO) ₂ + ethanol	0.5	0.5	650	Electrolyte optimization; Surface film adjustment
[28]	ZnSO ₄ +methanol	1	0.5	900	Electrolyte optimization
[29]	ZnSO ₄ +sorbitol	1	1	1000	Electrolyte optimization; Orientation adjustment
[30]	ZnSO ₄ + PG	2	2	1000	Electrolyte optimization; Orientation adjustment
[31]	ZnSO ₄ +tripropylene glycol	1	0.2	1100	Electrolyte optimization; Orientation adjustment
[32]	ZnSO ₄ + EG	0.5	0.5	1300	Electrolyte optimization; Orientation adjustment
[33]	Zn(CF ₃ SO ₃) ₂ +PEG	0.25	0.125	1500	Electrolyte optimization; Orientation adjustment
[34]	ZnSO ₄ + inositol	1	1	1700	Electrolyte optimization; Orientation adjustment
[35]	ZnSO ₄ +sorbitol	1	0.5	2000	Electrolyte optimization; Surface adjustment
This work	ZnSO ₄ +ZnI ₂ + n -butanol	1	1	3000	Electrolyte optimization; Orientation adjustment; Surface film adjustment

The corresponding references are listed in the supporting information

5. Authors should explain why 3% v/v *n*-butanol is used in the electrolyte. Have experiments been done with other percentages of *n*-butanol? How does the amount of *n*-butanol affect the battery?

Response: We appreciate your comment. We have conducted experiments with different concentration of *n*-butanol and determined that the 3% v/v concentration is optimal. We have included the relevant data and discussions in the revised manuscript and supporting information (Note 3, Supplementary Figs. 13-15). In summary, insufficient *n*-butanol does not effectively mitigate the I₃⁻ shuttle, while excessive *n*-butanol interferes with the participation of I₃⁻ in the reaction. Therefore, a moderate concentration of *n*-butanol ensures efficient conversion reactions while minimizing iodine loss, which is why we selected 3% v/v as the optimal concentration.

Changes:

and the optimal amount (3% v/v) of *n*-butanol is determined through experiments (the corresponding discussion and data are included in the Note 3 and Supplementary Figs. 13-15). (Page 13 of the revised Manuscript)

Supplementary Fig. 13 Rate performance comparison of ZSI-*n*-1%, ZSI-*n*-3% and ZSI-*n*-6%. (Page 14 of the revised SI)

Supplementary Fig. 14 Capacity and Coulombic efficiency of ZSI-*n*-1%, ZSI-*n*-3% and ZSI-*n*-6% at 5 A g⁻¹. (Page 15 of the revised SI)

Supplementary Fig. 15 Photos of (a) ZSI-*n*-1%, (b) ZSI-*n*-3% and (c) ZSI-*n*-6% systems in fully charge state. (The left electrode is cathode and the right electrode is anode in these two photos) (Page 16 of the revised SI)

Note 3: To find the optimal amount, different concentration of *n*-butanol (1% v/v, 3% v/v, 6% v/v) is incorporated into the ZSI electrolyte, which are named as ZSI-*n*-1%, ZSI-*n*-3%, and ZSI-*n*-6%, respectively. As shown in Supplementary Fig. 13, the full cell with the ZSI-*n*-3% electrolyte exhibits a significantly improved specific capacity, reaching 0.47 mAh cm⁻² at 0.5 A g⁻¹. At the same current density, the specific capacities of cells with ZSI-*n*-1% and ZSI-*n*-6% are 0.34 mAh cm⁻² and 0.45 mAh cm⁻², respectively, both lower than that of ZSI-*n*-3%. Furthermore, at a high current density of 10 A g⁻¹, the full cell with ZSI-*n*-3% maintains a specific capacity of 0.30 mAh cm⁻², significantly higher than those with ZSI-*n*-1% (0.03 mAh cm⁻²) and ZSI-*n*-6% (0.26 mAh cm⁻²).

More importantly, the Zn-I₂ battery with ZSI-*n*-3% demonstrates excellent reversible capacity of 0.30 mAh cm⁻² after 30,000 cycles at a high current density of 5 A g⁻¹, with a low-capacity decay rate of approximately 0.0012% per cycle (Supplementary Fig. 14). In contrast, the battery with ZSI-*n*-1% begins to decay immediately, likely due to insufficient *n*-butanol to limit the I₃⁻ shuttle effect, resulting in significant iodine loss (as shown in the left image of Supplementary Fig. 15, where the I₃⁻ shuttle effect is not fully suppressed). Conversely, the battery with ZSI-*n*-6% initially maintains good stability over the first 3,500 cycles with minimal capacity degradation, likely due to the high amount of *n*-butanol effectively fixing I₃⁻ near the cathode. However, in subsequent cycles, the battery shows significant performance decline, with capacity approaching zero by the end of 10,000 cycles. This decline may be attributed to excessive *n*-butanol accumulation on the cathode surface, which impedes the participation of the outer layer of I₃⁻ in the reaction, leading to instability in the conversion reaction (evident from the right image of Supplementary Fig. 15, where I₃⁻ is entirely fixed at the cathode electrode by *n*-butanol).

In summary, too little *n*-butanol fails to adequately mitigate the I₃⁻ shuttle effect, while excessive *n*-butanol hinders I₃⁻ participation in the reaction. Therefore, a moderate concentration of *n*-butanol ensures efficient conversion reactions while minimizing iodine loss, which is why we selected 3% v/v as the optimal amount. (Page 49 of the revised SI)

6. The author mentioned in the manuscript that ZSI-*n* is a suspended electrolyte. If the author provides additional optical photos of three electrolytes, it will be beneficial for readers to understand.

Response: Thanks for the valuable comment. We have supplemented the optical microscopy images of these three electrolytes in Supplementary Fig. 40. It can be seen that ZSI-*n* electrolyte (Supplementary Fig. 40c) exhibits many small droplets in the optical microscope, while ZS (Supplementary Fig. 40a) and ZSI (Supplementary Fig. 40b) electrolytes present a homogeneous solution. Moreover, we also provided more intuitive photos and added them to the Supplementary Fig. 41. ZSI-*n* electrolyte appears opaque, while ZS and ZSI electrolytes appear clear and transparent. These results indicate that ZSI-*n* is a suspended electrolyte. In addition, we have added the corresponding discussion in the manuscript.

Changes:

Furthermore, the photos of three electrolytes are shown in the Supplementary Fig. 40. Where the ZSI-*n* electrolyte appears opaque, while ZS and ZSI electrolytes appear clear. In addition, optical microscope was used to further characterize the electrolytes. As shown in Supplementary Fig. 41c, there are many small droplets in ZSI-*n* electrolyte, while ZS (Supplementary Fig. 41a) and ZSI (Supplementary Fig. 41b) electrolytes present a homogeneous solution. These results indicate that ZSI-*n* is a suspended electrolyte due to the low solubility of *n*-butanol in aqueous solution. Thus, SEI film containing *n*-butanol has the robust hydrophobicity, which effectively prevents water molecules from contacting the zinc surface, thereby suppressing unpleasant side reactions⁴⁸. (Page 23 of the revised Manuscript)

Supplementary Fig. 40 The pictures of ZS, ZSI and ZSI-*n* electrolytes. (Page 41 of the revised SI)

Supplementary Fig. 41 Optical microscopy images of (a) ZS, (b) ZSI and (c) ZSI-*n* electrolytes. (Page 42 of the revised SI)

7. *The author should elaborate on the significance of the increased Zn^{2+} transference number during the zinc deposition/stripping process, which will provide crucial insights into the mechanism behind the battery's improved performance and long cycle life*

Response: Thanks for your value comment. The increase in Zn^{2+} migration number plays a crucial role in improving the electrochemical performance of zinc anodes. It can reduce the concentration changes during zinc deposition/stripping, thereby leading to more uniform zinc deposition. (*Adv. Funct. Mater.* **2020**, *30*, 2001317; *Adv. Energy Mater.* **2021**, *11*, 2003065).

According to your suggestion, we have added a discussion section to the main text.

Changes:

The increase in the migration number of Zn^{2+} reduces the Zn^{2+} concentration changes during zinc deposition/stripping, thereby making zinc deposition more uniform^{40,41}. (Page 19 of the revised Manuscript)

8. *The author analyzed the changes in solvation structure through theoretical calculations and structural characterization, but did not explain the effects of solvation structure. It is better to added the corresponding discussion in the “Characterization and Simulation of Electricity Structure” section.*

Response: We thanks for the valuable comment. Regarding the effect of changes in solvation structure, we have conducted corresponding tests and discussions in the

manuscript already.

1. Reduce the overpotential

The discussions in the manuscript: As shown in Fig. 4a, the overpotential of ZSI-*n* reaches an exceptionally low value (11.2 mV), while the overpotential of ZSI is slightly higher (22.1 mV), but still lower to that of ZS (92.0 mV). Besides, a similar decreasing trend of overpotential is observed in CV tests on Zn//Zn symmetric cells (Supplementary Fig. 24) (Pages 17-18 of the revised Manuscript)

2. Increase the Zn²⁺ transference number

The discussions in the manuscript: The Zn²⁺ transference number ($t_{Zn^{2+}}$) of ZIS-*n* shown in Supplementary Fig. 28 is calculated to be 0.38, higher than that of ZSI (0.33) and ZS (0.30)³⁹. The increase in the migration number of Zn²⁺ reduces the Zn²⁺ concentration changes during zinc deposition/stripping, thereby making zinc deposition more uniform^{40,41}. (Page 19 of the revised Manuscript)

3. Reduce the activation energy

The discussions in the manuscript: Moreover, as shown in Supplementary Fig. 29-30, the E_a values for ZSI-*n*, ZSI, and ZS are 36.01, 41.96, and 56.46 kJ mol⁻¹, respectively. The lower E_a indicates an enhanced dynamic process for the electrocatalytic redox reactions of iodine^{42,43}. (Page 19 of the revised Manuscript)

4. Form a SEI film and inhibit zinc dendritic growth

The discussions in the manuscript: Furthermore, the ZSI-*n* curve exhibits two new peaks in Supplementary Fig. 37b at 31.73° and 34.37°, corresponding to the presence of zinc oxide (ZnO, JCPDS: 89-1397). Meanwhile, another new peak of ZSI-*n* at 12.82° indicates the generation of zinc carbonate ($Zn_4CO_3(OH)_6 \cdot H_2O$, JCPDS: 11-0287), presenting trace amounts on ZS and ZSI (Supplementary Fig. 37c). The presence of Zn-O bonds in the Zn spectrum (Supplementary Fig. 38b) and the increased ZnCO₃ peak in the C spectrum (Supplementary Fig. 38c) of the ZSI-*n* curve confirm the enrichment of ZnO and $Zn_4CO_3(OH)_6 \cdot H_2O$ on Zn surface. Additionally, high-resolution transmission electron microscopy (HRTEM) demonstrates that a solid electrolyte interface (SEI) layer is composed of the organic phase and the inorganic phase consisting of ZnO and $Zn_4CO_3(OH)_6 \cdot H_2O$, as evidenced by the unique lattice fringe of

both observed in the SEI (Fig. 4g). The formation of the organic phase is supported by the strong adsorption capacity of *n*-butanol on the zinc surface, as revealed by DFT calculations in Fig. 4i. The adsorption energies on Zn (002) for various particles, including *n*-butanol, SO_4^{2-} , H_2O , and I^- , exhibit significant differences (-72.24, -25.93, -12.21, and -1.99 eV, respectively). (Pages 22-23 of the revised Manuscript)

To further elucidate the morphology of zinc deposition, SEM tests were employed to examine the surface of zinc after plating (Supplementary Fig. 35). The inhomogeneous distribution of large zinc dendrites is formed on the zinc surface in ZS electrolyte, but the zinc surface in ZSI is uniform and even. Furthermore, the inclusion of *n*-butanol in ZSI results in a tighter and flatter surface, effectively suppressing dendrite growth and extending the lifespan of the battery. The substantial difference in surface characteristics was more pronounced in a laser scanning confocal microscope (LSCM), as shown in Fig. 4h and Supplementary Fig. 36a-b. A pronounced longitudinal drop is observed on the zinc surface in ZS, suggesting the growth of numerous large-sized and uneven dendrites. (Page 21 of the revised Manuscript)

Fig. 4a Voltage profiles of galvanostatic Zn deposition on Cu at 1 mA cm^{-2} .

Supplementary Fig. 28 Zn^{2+} transference number ($t_{\text{Zn}^{2+}}$) of the symmetric cell with (a)

ZS, (b) ZSI and (c) ZSI-*n* electrolytes.

Supplementary Fig. 30 Charge transfer resistances for varying temperature of (a) ZS, (b) ZSI and (c) ZSI-*n* electrolytes.

Fig. 4g High-resolution TEM images of the Zn anode surface after cycling in three electrolytes.

Fig. 4i Values of surface adsorption energy of H_2O , SO_4^{2-} , I^- and *n*-butanol on Zn (002) plane.

Fig. 4f Optical microscopy image of Zn/electrolyte interface during Zn deposition in ZS, ZSI and ZSI-*n*.

Supplementary Fig. 35 SEM images of Zn anode after 50 cycles in the (a) ZS, (b) ZSI and (c) ZSI-*n* electrolytes.

Fig. 4h LCSM images of Zn anodes after cycling in ZSI-*n*. and **Supplementary Fig. 36** LCSM images of Zn anodes after cycling in **b** ZS and **c** ZSI electrolytes.

9. The performance of the prepared Zn-I₂ battery appears to be good, but there is a lack of comparative data with similar types of batteries. It is recommended to add a performance comparison table with other latest reports.

Response: We appreciate your valuable comments. The team have conducted

comprehensive reporting on Zn-I₂ batteries over the past three years (Table 1). It can be seen that our work ranks among the top in terms of mass loading, cycle life and area capacity.

Change:

Table 1. Summary of the optimization strategy, load mass, current density, area capacity density, area energy density and cycle number of Zn-I₂ batteries reported recently. (Page 45 of the revised SI)

Reference	Optimization strategy	Load mass (mg cm ⁻²)	Current density (A g ⁻¹)	Area capacity density (mAh cm ⁻²)	Area energy density (mWh cm ⁻²)	Cycle number
[1]	Cathode	0.06	1	0.014	0.016	1500
[2]	Cathode Anode	1.0	2.5	0.118	0.13	3000
[3]	Cathode	1.0	2.11	0.175	0.21	10000
[4]	Anode	0.63	2	0.126	0.15	10000
[5]	Electrolyte additive In-situ SEI	1.38	1.45	0.259	0.30	5000
[6]	Cathode	1.0	5	0.095	0.11	20000
[7]	Anode	1.3	2	0.135	0.15	20000
[8]	Anode	1.0-2.0	3.2	0.09-0.17	0.10-0.20	6000
[9]	Cathode	1.0	1	0.2	0.25	5700
[10]	Cathode	1.0	4.22	0.14	0.16	10000
[11]	Cathode	0.8-1.5	2	0.16-0.3	0.19-0.35	10000
[12]	Cathode	1.25	4	0.25	0.28	1500
[13]	Gel electrolyte	1.5	1.58	0.24	0.26	5000
[14]	Cathode	2	5	0.24	0.29	5000
[15]	Cathode	2	8.44	0.16	0.19	6000
[16]	Cathode	1.0-1.2	2.11	0.15-0.18	0.16-0.20	20000
[17]	Cathode	1.17	4.22	0.154	0.20	5000
[18]	Cathode	1.0-1.2	1.2	0.21-0.25	0.27-0.32	1000
[19]	Cathode	1.0-1.2	5	0.16-0.19	0.19-0.22	30000
[20]	Gel electrolyte	0.8-1.0	2.11	0.12-0.15	0.14-0.17	22000
[21]	Ionic liquid electrolyte	0.8-1.0	4	0.12-0.15	0.14-0.17	18000
[22]	Cathode	1.0-1.2	5	0.17-0.2	0.20-0.24	17000

[23]	Cathode	1.0-2.0	2	0.1-0.2	0.12-0.23	1400
[24]	Cathode	0.8-1.2	3	0.1-0.15	0.12-0.18	5900
This work	Cathode Electrolyte additive In-situ SEI	1.45	5	0.30	0.34	30000

The corresponding references are listed in the supporting information

Point-by-Point Response to Referee Reports

Reviewer #1 (Remarks to the Author):

I sincerely appreciate the authors' thorough explanations and the additional data provided. However, I must express disappointment regarding the error in the ZnI₂ concentration, as this is a critical factor for the experiments and is important for readers to assess the study accurately.

*Furthermore, the authors state, "the mass loading of I₂ and the areal energy density of the battery are as high as 1.45 mg cm⁻² and 0.34 mWh cm⁻², respectively, which are higher than most current Zn-I₂ batteries." However, a high-loading cathode (37.5 mg iodine cm⁻²) has already been achieved and published in *Advanced Materials* (2024, 36, 2404011). It is not reasonable to claim that this lower performance merits publication in the high-quality journal *Nature Communications*.*

Response: We would like to thank you for your critical comments.

Firstly, we profoundly apologize for the inconvenience arising from the inaccuracy in the concentration. In the process of optimizing the electrolyte concentration, different concentrations of ZnI₂ from 0.02 M to 2.0 M were tested, and the wrong concentration was written accidentally in the initial manuscript. To rectify this and uphold the integrity of our findings, we have meticulously reviewed all parameters within the manuscript and appended original data sources, thereby reinforcing the reliability and authenticity of our data.

Secondly, you criticized our work for having lower mass loading compared to the recent result published in *Advanced Materials* (2024, 36, 2404011). We appreciate you bringing this to our attention. In response, we have prepared additional batteries with higher I₂ loading by adjusting the electrolyte concentration. As illustrated in the Supplementary Fig. 43, the mass loading of I₂ reaches 38.5 mg cm⁻², which is the highest value reported so far, also exceeded that (37.5 mg cm⁻²) of the research (*Advanced Materials*, 2024, 36, 2404011). More importantly, while with such a high mass loading, the battery exhibits a specific capacity of 125.9 mAh g⁻¹, which is also

superior to the performance of batteries in the Advanced Materials paper (83.3 mAh g⁻¹) and most other high-loading batteries. In addition, we have compiled a comprehensive overview on mass loading, capacity decay and cycle number of Zn-I₂ batteries published in recent years (as shown in Table R1). In this way, our work outperforms most other reports on Zn-I₂ batteries in terms of mass loading, capacity and cycle life. According, we have added these data and discussions to the revised manuscript and supporting information (Supplementary Fig. 43).

Table R1. Summary of load mass, Capacity decay every cycle, and cycle number of Zn-I₂ batteries reported recently.

Reference	Load mass (mg cm ⁻²)	Capacity decay every cycle (%)	Cycle number
[1]	1.0-1.2	0.002	4000
[2]	1.0-2.0	0.018	1500
[3]	1.25	0.0043	3500
[4]	1.45	0.004	3000
[5]	1.4	0.0013	5000
[6]	2.0	0.0015	5000
[7]	<2.1	0.0051	5000
[8]	1.0	0.0008	5700
[9]	<1.4	0.0044	5900
[10]	2.0	0.0025	6000
[11]	1.0	0.001	10000
[12]	0.63	0.002	10000
[13]	0.8-1.5	0.0026	10000
[14]	0.36-0.48	0.001	15000
[15]	0.75-1.1	0.0004	15000
[16]	1.0-1.2	0.0007	17000
[17]	0.8-1.0	0.001	18000
[18]	0.8-1.2	0.0034	20000
[19]	1.3	0.001	20000
[20]	1.0-1.2	0.0007	20000
[21]	0.8-1.0	0.0076	22000

[22]	1.0-2.0	0.0005	30000
[23]	37.5	0.005	1000
This work	1.45	0.0004	30000

Changes:

In addition, it should be noted that the mass loading of I₂ on the cathode can be adjusted by changing the electrolyte concentration in this study. As shown in Supplementary Fig. 43, the mass loading of I₂ reaches 38.5 mg cm⁻², which is the highest value reported so far⁴⁹. More importantly, while with such a high mass loading, the battery exhibits a specific capacity of 125.9 mAh g⁻¹. (Page 25 of the revised Manuscript)

Supplementary Fig. 43 Performance of specific capacity with different mass loading. (Page 44 of the revised SI)

Reference:

1. Wang, S. et al. Halide Exchange in Perovskites Enables Bromine/Iodine Hybrid Cathodes for Highly Durable Zinc Ion Batteries. *Adv. Mater.* **36**, 2401924 (2024).
2. Yang, J. et al. Janus Binder Chemistry for Synchronous Enhancement of Iodine Species Adsorption and Redox Kinetics toward Sustainable Aqueous Zn-I₂ Batteries. *J. Am. Chem. Soc.* **146**, 6628-6637 (2024).
3. Gao, W. et al. Efficient Charge Storage in Zinc-Iodine Batteries based on Pre-Embedded Iodine-Ions with Reduced Electrochemical Reaction Barrier and Suppression of Polyiodide Self-Shuttle Effect. *Adv. Funct. Mater.* **33**, 2211979 (2023).

4. Yan, L. et al. Multifunctional porous carbon strategy assisting high performance aqueous zinc-iodine battery. *Carbon*, **187**, 145-152 (2022).
5. Chen, S. et al. Interface Coordination Stabilizing Reversible Redox of Zinc for High-Performance Zinc-Iodine Batteries. *Small*, **18**, 2200168 (2022).
6. Yang, F. et al. Single atom catalysts for triiodide adsorption and fast conversion for boosted performance in aqueous zinc-iodine batteries. *Energy Environ. Sci.* **16**, 4630-4640 (2023).
7. Guo, C. et al. Cobalt Single-Atom Electrocatalysts Enhanced by Hydrogen-Bonded Organic Frameworks for Long-Lasting Zinc-Iodine Batteries. *Adv. Funct. Mater.* **34**, 2314189 (2024).
8. Wang, S. et al. Conversion-Type Organic-Inorganic Tin-Based Perovskite Cathodes for Durable Aqueous Zinc-Iodine Batteries. *Adv. Energy Mater.* **13**, 2300922 (2023).
9. Qu, W. et al. Ni Single-Atom Bual Catalytic Electrodes for Long Life and High Energy Efficiency Zinc-Iodine Batteries. *Small*, **20**, 2310475 (2024).
10. Sun, J. et al. Heavily heteroatoms doped carbons with tunable microstructure as the iodine hosts for rechargeable zinc-iodine aqueous batteries. *J. Alloys Compd.* **947**, 169696 (2023).
11. Liu, M. et al. Physicochemical Confinement Effect Enables High-Performing Zinc-Iodine Batteries. *J. Am. Chem. Soc.* **144**, 21683-21691 (2022).
12. Peng, H. et al. Constructing fast-ion-conductive disordered interphase for high-performance zinc-ion and zinc-iodine batteries. *Matter*, **5**, 1-16 (2022).
13. Wei, F. et al. 2D Mesoporous Naphthalene-Based Conductive Heteroarchitectures toward Long-Life, High-Capacity Zinc-Iodine Batteries. *Adv. Funct. Mater.* **34**, 2310693 (2023).
14. Xu, Z, et al. Crowding Effect-Induced Zinc-Enriched/Water-Lean Polymer Interfacial Layer Toward Practical Zn-Iodine Batteries. *ACS Nano*, **17**, 23207-23219 (2023).
15. He, J. et al. Synergistic Effect of Lewis Acid-Base and Coulombic Interactions for High-performance Zn-I₂ Batteries. *Energy Environ. Sci.* **17**, 323-331 (2024).
16. Li, Y. et al. Built-in electrocatalytic nanoreactors anchoring ultrahigh iodine utilization for long-lasting zinc-iodine batteries. *Chemical Engineering Journal*, **483**,

149320 (2024).

17. Xiao, T. et al. All-Round Ionic Liquids for Shuttle-Free Zinc-Iodine Battery. *Angew. Chem. Int. Ed.* **63**, e202318470 (2024).

18. Wei, F. et al. Mesoporous Poly (3,4-ethylenedioxythiophene): Poly (styrenesulfonate) as Efficient Iodine Host for High-Performance Zinc-Iodine Batteries. *ACS Nano*, **17**, 20643-20653 (2023).

19. Wang, G. et al. In situ Construction of Multifunctional Surface Coatings on Zinc Metal for Advanced Aqueous Zinc-Iodine Batteries. *Adv. Energy Mater.* **14**, 2303221 (2024).

20. Chen, Q. et al. Synergic anchoring of Fe₂N nanoclusters on porous carbon to enhance reversible conversion of iodine for high-temperature zinc-iodine battery. *Nano Energy*, **117**, 108897 (2023).

21. Yang, J. et al. Cation-Conduction Dominated Hydrogels for Durable Zinc-Iodine Batteries. *Adv. Mater.* **36**, 2313610 (2024).

22. Du, Y. et al. Lithiation Enhances Electrocatalytic Iodine Conversion and Polyiodide Confinement in Iodine Host for Zinc-Iodine Batteries. *Adv. Funct. Mater.* **33**, 2304811 (2023).

23. Zhang, S. et al. Protein Interfacial Gelation toward Shuttle-Free and Dendrite-Free Zn-Iodine Batteries. *Adv. Mater.* 2404011 (2024).

Reviewer #2(Remarks to the Author):

All concerns have been addressed effectively. and I advocate for the publication of this manuscript.

Response: We sincerely thank the reviewer #2 for his/her dedication and recognition of our work, and we are also grateful for his/her professional and meticulous insights, which have improved the quality of our manuscript.

Reviewer #3 (Remarks to the Author):

The manuscript has been improved, and it can be accepted now.

Response: We sincerely thank the reviewer #3 for his/her hard work and positive evaluation of our work.